# EXPLOITING REDUNDANCY: SEPARABLE GROUP CONVOLUTIONAL NETWORKS ON LIE GROUPS

## ABSTRACT

Group convolutional neural networks (G-CNNs) have been shown to increase parameter efficiency and model accuracy by incorporating geometric inductive biases. In this work, we investigate the properties of representations learned by regular G-CNNs, and show considerable parameter redundancy in group convolution kernels. This finding motivates further weight-tying by sharing convolution kernels over subgroups. To this end, we introduce convolution kernels that are separable over the subgroup and channel dimensions. In order to obtain equivariance to arbitrary affine Lie groups we provide a continuous parameterisation of separable convolution kernels. We evaluate our approach across several vision datasets, and show that our weight sharing leads to improved performance and computational efficiency. In many settings, separable G-CNNs outperform their non-separable counterpart, while only using a fraction of their training time. In addition, thanks to the increase in computational efficiency, we are able to implement G-CNNs equivariant to the $\text{Sim}(2)$ group; the group of dilations, rotations and translations of the plane. $\text{Sim}(2)$-equivariance further improves performance on all tasks considered, and achieves state-of-the-art performance on rotated MNIST.

## 1 INTRODUCTION

Minsky & Papert (1988) suggest that the power of the perceptron comes from its ability to *learn to discard* irrelevant information. In other words; information that does not bear significance to the current task does not influence representations built by the network. According to Minsky & Papert (1988), this leads to a definition of perceptrons in terms of the *symmetry groups* their learned representations are invariant to. Progress in geometric deep learning has shown the power of pro-actively equipping models with such geometric structure as inductive bias, reducing model complexity and improving generalisation and performance (Bronstein et al., 2017). An early example of such geometric inductive bias at work can be seen in the convolutional layer in a CNN (LeCun et al., 1998). CNNs have been instrumental in conquering computer vision tasks, and much of their success has been attributed to their use of the convolution operator, which commutes with the action of the translation group. This property, known as *equivariance* to translation, comes about as a result of the application of the same convolution kernel throughout an input signal, enabling the CNN to learn to detect the same features at any location in the input signal, directly exploiting translational symmetries that naturally occur in many tasks.

Although invariance to object-identity preserving transformations has long been recognised as a desirable model characteristic in machine learning literature (Kondor, 2008; Cohen, 2013; Sifre & Mallat, 2014), only recently Cohen & Welling (2016a) introduced the Group Equivariant CNN (G-CNN) as a natural extension of the CNN (LeCun et al., 1998), generalising its equivariance properties to group actions beyond translation. The layers of a G-CNN are explicitly designed to be equivariant to such transformations, hence the model is no longer burdened with *learning* invariance to transformations that leave object identity intact. It has since been shown that equivariant deep learning approaches may serve as a solution in fields that as of yet remain inaccessible to machine learning due to scarce availability of labelled data, or when compact model design due to limited computational power is required (Winkels & Cohen, 2018; Linmans et al., 2018; Bekkers, 2019).

**Complexity and redundancy issues impeding regular group convolutions** A growing body of work shows applications of G-CNNs consistently and decisively outperforming classical CNNs

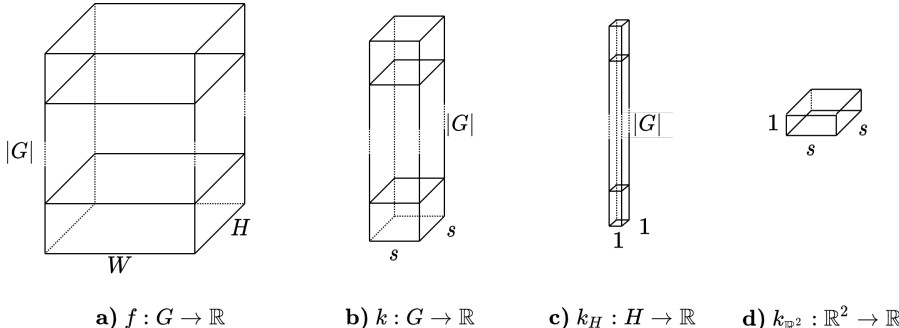

**a)** $f : G \to \mathbb{R}$    **b)** $k : G \to \mathbb{R}$    **c)** $k_H : H \to \mathbb{R}$    **d)** $k_{\mathbb{R}^2} : \mathbb{R}^2 \to \mathbb{R}$

Figure 1: In group convolutions on affine Lie groups, a feature map $f$ defined over the group, depicted in Fig. **a)** is convolved with a filter $k$ shown in Fig. **b)** of size $|G| \times s \times s$, where $s$ is the kernel size, and $|G|$ the size of the group. We propose factorising this convolution into a convolution over the group with a kernel $k_H$ shown in Fig. **c)** with size $|G| \times 1 \times 1$, followed by a convolution over the spatial dimensions with a kernel $k_{\mathbb{R}^2}$ shown in Fig. **d)** with size $1 \times s \times s$.

(Worrall et al., 2017; Weiler et al., 2018a; Bekkers et al., 2018; Esteves et al., 2018; Bekkers, 2019; Worrall & Welling, 2019; Sosnovik et al., 2021b). However, a practical challenge impeding application to larger groups is the computational complexity of regular group convolutions, which scales exponentially with the dimensionality of the group. Furthermore, Lengyel & van Gemert (2021) show that group convolution filters in the original formulation of the G-CNN by Cohen & Welling (2016a) exhibit considerable redundancies along the group axis for the $p4m$ and $\mathbb{Z}^2$ groups. Similar observations motivated depthwise separable convolutions (Chollet, 2017), which not only increased parameter efficiency but also model performance; observed correlations between weights are explicitly enforced with further parameter sharing through the use of kernels separable along spatial and channel dimensions. We address the observations of redundancy along with the scalability issues of regular G-CNNs in their current form. Our paper contains the following contributions:

- We introduce separable group convolutions for affine Lie groups $\mathbb{R}^n \rtimes H$, sharing the kernels for translation elements $x \in \mathbb{R}^n$ along subgroup elements $h \in H$. See Fig. 1 for an overview.

- We propose the use of a SIREN (Sitzmann et al., 2020) as kernel parameterisation in the Lie algebra - imposing a fixed number of parameters per convolution kernel, regardless of the resolution at which this kernel is sampled, and ensuring smoothness over the Lie group.

- Separable group convolutions allow us to build $\mathrm{Sim}(2)$-CNNs, which we thoroughly experiment with. We show equivariance to $\mathrm{Sim}(2)$ increases accuracy over a range of vision benchmarks.

- To achieve equivariance to continuous affine Lie groups, we propose a random sampling method over subgroups $H$ for approximating the group convolution operation.

First, we position this work within the area of equivariant deep learning by giving an overview of related works, and explaining which current issues we are addressing with this work. We derive separable group convolutions, and show how they may be applied to continuous groups. Lastly, we apply these ideas by experimenting with implementations for roto-translations in 2D ($\mathrm{SE}(2)$), dilation and translation in 2D ($\mathbb{R}^2 \rtimes \mathbb{R}^+$) and dilation, rotation and translation in 2D ($\mathrm{Sim}(2)$).

## 2    RELATED WORK

**Group equivariant convolutional neural networks** Broadly speaking, research on G-CNNs can be divided into two approaches. First, *Regular* G-CNNs use the left-regular representation of the group of interest to learn representations of scalar functions over the group manifold, or a quotient space of the group. The left-regular representation acts on the convolution kernels, yielding an orbit of the kernel under the group action. Convolving the input using these transformed filters, a feature map defined over the group is obtained at each layer. This approach most naturally extends the conventional CNN, where convolution kernels are transformed under elements of the translation group. Regular G-CNNs have been implemented for discrete groups (Cohen & Welling, 2016a; Winkels & Cohen, 2018; Worrall & Brostow, 2018), compact continuous groups (Marcos et al., 2017; Bekkers et al., 2018) and arbitrary non-compact continuous Lie groups (Bekkers, 2019; Finzi et al., 2020; Romero et al., 2020). However, practical implementations for continuous groups often require

some form of discretisation of the group, possibly introducing discretisation artefacts, and requiring a choice of *resolution* over the group. For the second class, *steerable* G-CNNs, representation theory is used to compute a basis of equivariant functions for a given group, which are subsequently used to parameterise convolution kernels (Cohen & Welling, 2016b; Weiler et al., 2018a;b; Sosnovik et al., 2019; 2021a). Although steerable G-CNNs decouple the cardinality of the group from the dimensionality of the feature maps, this approach is only compatible with compact groups.

The current paper may, in approach, be compared to Bekkers (2019) and Finzi et al. (2020), who define convolution kernels on the Lie algebra of continuous groups to enable convolutions on their manifold. Similarly, we make use of the Lie algebra and exponential map to obtain convolution kernels on the group, but separate the kernels by subgroups.

Bekkers (2019) defines a set of basis vectors in the Lie algebra, which, when combined with the exponential map, allow for the identification of group elements by a vector in $\mathbb{R}^n$. Subsequently, a set of B-splines is defined on the algebra, which form a basis to expand convolution kernels in. A linear combination of these bases creates a locally continuous function on the Lie algebra defining a convolution kernel and its behaviour under transformations of the group. Although this method allows for direct control over kernel smoothness, the learned convolution filters are limited in their expressivity by their basis functions. Finzi et al. (2020) instead use an MLP to learn convolution kernels on the Lie algebra, which in addition allows them to handle point cloud data. The MLP is constructed to learn kernel values at (arbitrary) relative offsets in the Lie algebra. In contrast, we propose to use SIRENs (Sitzmann et al., 2020) to parameterise convolution kernels, as they have been shown to outperform other forms of MLPs in parameterising convolution kernels (Romero et al., 2021), and offer more explicit control over kernel smoothness: a desirable property for addressing discretisation artefacts that occur when modelling features on continuous groups (see App. A.6).

**Separable filters in machine learning** In image processing, spatially separable filters have long been known to increase parameter- and computational efficiency, and learning such constrained filters may even increase model performance (Rigamonti et al., 2013). In Sifre & Mallat (2014), authors investigate SE(2)-invariant feature learning through scattering convolutions, and propose separating the group convolution operation for affine groups into a cascade of two filterings, the first along the spatial dimensions $\mathbb{R}^n$, and the second along subgroup dimension $H$. From this, authors derive a separable approach to the convolution operation with learnable filters as used in CNNs. This formulation has since been named the *depthwise-separable* convolution (Chollet, 2017), a special case of the Network-In-Network principle (Lin et al., 2013) which forms the basis for the success of the Inception architectures (Szegedy et al., 2015). In depthwise separable convolutions, each input channel first gets convolved using a (set of) kernel(s) with limited spatial support. Afterwards, a 1x1 convolution is used to project the feature set detected in the input channels to the output space. Chollet (2017) speculates that the Inception architectures are successful due to the explicit separation of spatial and channel mapping functions, whereas in conventional CNNs, kernels are tasked with simultaneously mapping inter-channel and spatial correlations.

Haase & Amthor (2020) argue that the original formulation of the depthwise-separable convolution reinforces inter-kernel correlations, but does not in fact leverage intra-kernel correlations. Subsequently, they propose an inverse ordering of the operations given in depthwise-separable convolutions, sharing the same spatial kernel along the input channels, and show convincing results. Extending this investigation of learned convolution filters to the original G-CNN (Cohen & Welling, 2016a), Lengyel & van Gemert (2021) remark on the high degree of correlation found among filters along the rotation axis, and propose to share the same spatial kernel for every rotation feature map. We attempt to generalise this approach, proposing separable convolutions on arbitrary affine Lie groups.

## 3  BACKGROUND

In the following section we give the theoretical background for implementing group convolutions for arbitrary affine Lie groups. We assume familiarity with the basics of group theory and provide the relevant concepts in Appx. A.1 and Appx. A.4. For simplicity of notation, we initially assume that our input signal/feature map $f$ has a single channel.

**Lifting convolutions** To preserve information on the pose of features in the input, an equivariant convolution operation is achieved by *lifting* a function from the input space to (a homogeneous space of) the group. As we are interested in real signals, specifically image data living on $\mathbb{R}^2$, we assume the Lie group of interest $H$ is taken in semidirect product with the domain of our data; $G = \mathbb{R}^2 \rtimes H$.

In group convolutions, a given kernel is left-acted by all transformations in $G$, thereby generating a signal on the higher dimensional space $G$ instead of $\mathbb{R}^2$. Hence, the output feature maps disentangle poses through a domain expansion, e.g. positions plus rotations or scales. For a given group element $h \in H$, kernel $k$, and location $\vec{x}$ in the input domain $\mathbb{R}^2$, the lifting convolution is given by:

$$(f *_{\text{lifting}} k)(g) = \int_{\mathbb{R}^2} f(\tilde{\boldsymbol{x}}) k_h(\tilde{\boldsymbol{x}} - \boldsymbol{x}) \, \mathrm{d}\tilde{\boldsymbol{x}}. \tag{1}$$

where $g=(\boldsymbol{x}, h)$ and $k_h = \frac{1}{|\det h|} \mathcal{L}_h[k]$ is the kernel $k : \mathbb{R}^2 \to \mathbb{R}$ transformed via the action of group element $h$ via $\mathcal{L}_h[k](\boldsymbol{x}) := k(h^{-1}\boldsymbol{x})$, and with $\det h$ the determinant of the matrix representation of $h$ that acts on $\mathbb{R}^d$. The output of lifting convolutions yields a $G$-feature map with the original two spatial input dimensions ($\mathbb{R}^2$), and an additional group dimension ($H$). See Fig. 10.

**Group convolutions** Now that the data is lifted to the domain of the group, we continue with group convolutions in subsequent layers. Given a kernel $k$ (now a function on $G$), Haar measures $\mathrm{d}\tilde{g}$ and $\mathrm{d}\tilde{h}$ on the group $G$ and sub-group $H$ respectively, group convolutions are given by:

$$\begin{aligned}
(f *_{\text{group}} k)(g) &= \int_G f(\tilde{g}) k(g^{-1} \cdot \tilde{g}) \, \mathrm{d}\tilde{g} = \int_G f(\tilde{g}) \mathcal{L}_g k(\tilde{g}) \, \mathrm{d}\tilde{g} \\
&= \int_{\mathbb{R}^2} \int_H f(\tilde{\boldsymbol{x}}, \tilde{h}) \mathcal{L}_x \mathcal{L}_h k(\tilde{\boldsymbol{x}}, \tilde{h}) \frac{1}{|h|} \, \mathrm{d}\tilde{\boldsymbol{x}} \, \mathrm{d}\tilde{h} \\
&= \int_{\mathbb{R}^2} \int_H f(\tilde{\boldsymbol{x}}, \tilde{h}) k(h^{-1}(\tilde{\boldsymbol{x}} - \boldsymbol{x}), h^{-1} \cdot \tilde{h}) \frac{1}{|h|} \, \mathrm{d}\tilde{\boldsymbol{x}} \, \mathrm{d}\tilde{h}.
\end{aligned} \tag{2}$$

Evaluating this convolution for every group element $g \in G$, we again obtain a function defined on $G$. As we know $G=\mathbb{R}^2 \rtimes H$, we can factor this operation into a transformation of a kernel $k$ by a group element $h \in H$, $k_h = \mathcal{L}_h(k)$, followed by a convolution at every spatial location in $f$. See Fig. 11.

**Achieving invariance** Using lifting and group convolution operations, we can construct convolutional layers that co-vary with the action of the group and explicitly preserve pose information throughout the representations of the network. In most cases, we ultimately want a representation that is invariant to transformations of the input in order to achieve invariance to these identity-preserving transformations. This is achieved by aggregating the information at all group elements in a feature map with an operation invariant to the group action, e.g., max-, mean- or sum-projection. In practice, this is done after the last group convolution, and is followed by one or more fully connected layers.

## 4  SEPARABLE GROUP CONVOLUTIONS ON LIE GROUPS

**Redundancies in group convolution filters** Similar to Haase & Amthor (2020); Lengyel & van Gemert (2021) we investigate parameter efficiency of learned convolution kernels to motivate separable filters. We train an SE(2)-equivariant CNN on the Galaxy10 dataset (see Sec. 5 for experimental details) and analyse the resulting group convolution kernels. We apply PCA by treating the values of the group convolution kernel for each subgroup element $h \in H$ as distinct spatial kernels with $k \times k$ features. The ratio of variance explained by the first principle component gives an indication of the variability of the group convolution kernel along the subgroup axis. If the ratio of explained variance is high, the distinct spatial convolution kernels along the subgroup axis are well-characterised by a single shared kernel. In Fig. 2, results are shown for the group convolution layers in our SE(2)-CNN before and after training. We find that during the training process, redundancy along the subgroup axis increases considerably. This motivates sharing a single spatial kernel along the subgroup elements, which can be achieved by separating the group convolution operation into a convolution over the subgroup $H$, followed by a convolution over the spatial domain $\mathbb{R}^2$.

**Separable group convolutions** Let us assume that the convolution kernel $k : G \to \mathbb{R}$ in Eq. 2 is separable. That is, $k(g)=k_{\mathbb{R}^2}(\boldsymbol{x}) k_H(h)$. In this case, we can derive the factorised separable group convolution as (see Appx. A.2 for the full derivation):

$$\begin{aligned}
(f *_{group} k)(g) &= \int_{\mathbb{R}^2} \int_H f(\tilde{\boldsymbol{x}}, \tilde{h}) k(h^{-1}(\tilde{\boldsymbol{x}} - \boldsymbol{x}), h^{-1} \cdot \tilde{h}) \frac{1}{|h|} \, \mathrm{d}\tilde{\boldsymbol{x}} \, \mathrm{d}\tilde{h} \\
&= \int_{\mathbb{R}^2} \left[ \int_H f(\tilde{\boldsymbol{x}}, \tilde{h}) k_H(h^{-1} \cdot \tilde{h}) \, \mathrm{d}\tilde{h} \right] k_{\mathbb{R}^2}(h^{-1}(\tilde{\boldsymbol{x}} - \boldsymbol{x})) \frac{1}{|h|} \, \mathrm{d}\tilde{\boldsymbol{x}}.
\end{aligned} \tag{3}$$

Here $k_H$ is a convolution kernel over the group $H$, and $k_{\mathbb{R}^2}$ is a convolution kernel over the spatial domain $\mathbb{R}^2$. We set $k^{\varphi}=k_H^{\varphi} k_{\mathbb{R}^2}^{\varphi}$, with $k_H^{\varphi}$ constant along $\mathbb{R}^2$ and $k_{\mathbb{R}^2}^{\varphi}$ constant along $H$. This can

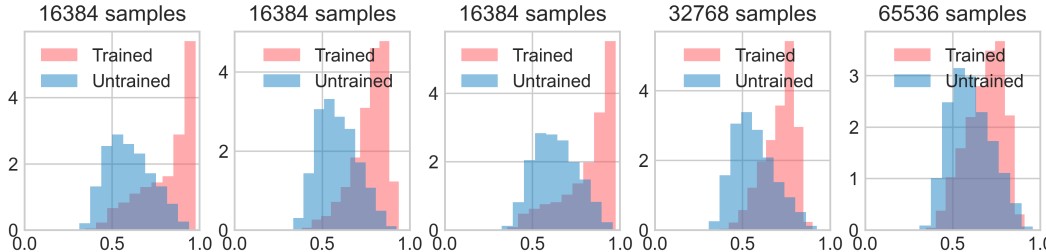

Figure 2: A set of histograms showing redundancy in learned group convolution kernels. On the x-axis is the ratio of variance explained by the first principal component when applying PCA on the set of spatial kernels along the group axis of a group convolution kernel. Y-axis shows the proportion of kernels with this explained variance ratio, where all bins sum to 1. The number of spatial kernels is listed in the title of each subfigure. Throughout the training process, redundancy in the group convolution kernels increases. Left to right: subsequent layers in the network.

be thought of as parameterising a convolution kernel $k$ over the group $G$ by sharing same spatial kernel $k^{\varphi}_{\mathbb{R}^2}$ weighed by a value $k^{\varphi}_H(h)$ at every input group element $h \in H$, see Fig. 12. Importantly, this factorisation greatly increases the efficiency of the group convolution operation. Since we can precompute the inner integral in Eq. 3, convolving over a single channel of a feature map $f$ of size $|H| \times x \times y$ goes from $O(|H|^2 \times x \times y \times k^2)$ to $O(|H| \times x \times y \times (|H| + k^2))$. See Fig. 13 for a visual intuition of the separable group convolution.

**Defining convolution kernels on Lie algebras** In order to perform lifting- and group convolutions (Eqs. 1, 2), we need to evaluate our convolution kernels $k$ at relative offsets $g'$ on the group. As in Bekkers et al. (2017); Weiler et al. (2018b); Bekkers (2019); Finzi et al. (2020), we express our group convolution kernel $k$ in analytical form, as a function of relative group elements yielding kernel values. Motivated by findings in Romero et al. (2021), we use a Sinusoidal Representation Network (SIREN) (Sitzmann et al., 2020) as kernel parameterisation. SIRENs lead to great performance improvements over MLPs with $\mathrm{ReLU}, \mathrm{LeakyReLU}$ and $\mathrm{Swish}$ when parameterising convolution kernels (we replicate this comparison for G-CNNs in Appx. C.4). Furthermore, the SIREN offers explicit control of kernel smoothness through a frequency multiplier parameter $\omega_0$: an important property discussed in Appx. A.6. Since the affine Lie group can be non-euclidean, and neural networks are functions generally defined over euclidean spaces, we resort to defining the kernel function on the Lie algebra of our group of interest (Finzi et al., 2020). The kernel function $k_\theta$ maps points in the Lie algebra (which may be associated with the relative offsets on the group we are convolving over by the exponential map) to kernel values, $k^{\varphi} : \mathfrak{g} \to \mathbb{R}$. For a given element $g$, we have: $k^{\varphi}(g) = \mathrm{SIREN}(\log g)$, see Fig. 14. The separable kernels follow the same principle, separated by subgroups, see Appx. B.1.

**Approximating equivariance for compact continuous and non-compact continuous groups** In the case of small discrete subgroups $H$, such as the group $C_4$ of rotations by 90 degrees, we are able to perform lifting- and group convolution operations exactly equivariant to the action of the group, as the integral given in Eq. 2 is tractable. We consider image data in particular, which is defined over a discrete grid $\mathbb{Z}^2$, a subgroup of $\mathbb{R}^2$, making both integrals over the group discrete sums:

$$(f *_{\mathrm{group}} k)(g) = \sum_{\tilde{\boldsymbol{x}} \in \mathbb{Z}^2} \sum_{\tilde{h} \in H} f(\tilde{\boldsymbol{x}}, \tilde{h}) k(\mathcal{T}_{h^{-1}}(\tilde{\boldsymbol{x}} - \boldsymbol{x}), h^{-1} \cdot \tilde{h}) \frac{1}{|h|} \Delta \tilde{\boldsymbol{x}} \Delta \tilde{h}, \tag{4}$$

with $\Delta \tilde{\boldsymbol{x}}$ and $\Delta \tilde{h}$ denoting the volume elements corresponding to the grid points. In the case of continuous groups, it is possible to either make a discretisation of the subgroup $H$, or to approximate the group convolution by means of random sampling. To obtain a volumetrically uniform sampling grid over the group, we sample a set of $n$ equidistant points in the Lie algebra as in Bekkers (2019), and map those to the group using the exponential map to obtain a grid $\mathcal{H} := [h_e, ..., h_n]$, see Fig. 15. For noncompact $H$ such as the dilation group $\mathbb{R}^+$, we localize the support in the Lie algebra. In the case of compact continuous groups such as $\mathrm{SO}(2)$ we approximate the integral by convolving over a uniformly spaced grid $\mathcal{H}$ which is perturbed by left-multiplication with a uniformly randomly sampled group element $h_\epsilon \sim \mathrm{d}\tilde{h}$, i.e., $h_\epsilon \mapsto \mathcal{H}_\epsilon := h_\epsilon \mathcal{H}$. By uniform sampling of $H$ we obtain left-invariant $\Delta \tilde{h}$ which only scales the overall convolution result, allowing us to omit it from 4. We further let $\Delta \tilde{\boldsymbol{x}} = 1$ and obtain the discrete separable group convolution for continuous groups:

$$(f *_{group} k)(g) \approx \sum_{\tilde{\boldsymbol{x}} \in \mathbb{Z}^2} \sum_{\tilde{h} \in \mathcal{H}_\epsilon} f(\tilde{\boldsymbol{x}}, \tilde{h}) k(h^{-1}(\tilde{\boldsymbol{x}} - \boldsymbol{x}), h^{-1} \cdot \tilde{h}) \frac{1}{|h|}$$

$$= \sum_{\tilde{\boldsymbol{x}} \in \mathbb{Z}^2} \Big[ \sum_{\tilde{h} \in \mathcal{H}_\epsilon} f(\tilde{\boldsymbol{x}}, \tilde{h}) k_H(h^{-1} \cdot \tilde{h}) \Big] \frac{1}{|h|} k_{\mathbb{R}^2}(h^{-1}(\tilde{\boldsymbol{x}} - \boldsymbol{x})). \quad (5)$$

During training and inference, randomly sampling over the rotation group yields an unbiased estimation (Wu et al., 2019), making the network approximately equivariant to the continuous group $SO(2)$. Discretisation makes the network exactly equivariant only to a discretised subgroup of $SO(2)$, biasing the network to be equivariant to a fixed subset of transformations. In our experiments we show that randomly sampling outperforms discretisation when modelling equivariance to $SO(2)$.

**Channel support of separable group convolution kernels** In CNNs (and G-CNNs), a feature map $f^l$ at layer $l$ generally has multiple channels $C^l$; $f : \mathbb{R}^2 \to \mathbb{R}^{C^l}$. Factorising the group convolution operation into distinct subgroup and spatial dimensions presents us with a choice: we can either (1) define both $k_H^\varphi$ and $k_{\mathbb{R}^2}^\varphi$ over the input channels, or choose to (2) reduce the support of our spatial kernel to a single channel, and share a reweighing of it along the input channels in an inverse depth-wise separable manner, similar to Haase & Amthor (2020). An ablation study detailed in Appx. C.2 shows that on a fixed parameter budget, additional depthwise separation (2) consistently outperforms (1) defining both kernels over the input channels. As such, we will keep to the additional depthwise separation and refer to this implementation as the separable group convolution.

**Expressivity of separable group convolutions** Separable convolution kernels are strictly less expressive than their non-separable counterpart, but to what extent would this limit the expressivity of their learned representations? In separable group convolutions, we are sharing a weighted version of a single spatial kernel along the input group axis. In contrast, we could view non-separable group convolutions as having the ability to learn distinct spatial configurations of features along the group axis. For a visual example, see Appx. A.5. Although this reduction in expressivity could theoretically prove limiting in the application of G-CNNs on vision problems, in the next section we experimentally show that separable group convolutions in fact often outperform their non-separable counterpart.

## 5 EXPERIMENTS

We empirically motivate the use of seperable group convolutions by studying the performance of three G-CNNs that incorporate equivariance to three distinct groups. As our goal is to isolate the effect of separating the group convolution, we use the same shallow ResNet architecture throughout all experiments, only varying the sampling resolution over the group, see Appx. B.1 for details. After the last convolution block, we apply max-projection over the remaining group dimensions to achieve an invariant representation. We experiment with three different groups acting on $\mathbb{R}^2$: the roto-translation group ($SE(2)$), the translation-dilation group ($\mathbb{R}^2 \rtimes \mathbb{R}^+$) and the group of rotations, dilations and translations ($Sim(2)$). For $SE(2)$, we first evaluate the influence of random sampling versus discretisation of $SO(2)$. Next, we assess the difference in performance of separable and non-separable group convolutions on transformed MNIST variants; MNIST-rot, MNIST-scale for $SE(2)$ and $\mathbb{R}^2 \rtimes \mathbb{R}^+$ respectively. Due to the reduction in computational complexity separable group convolutions bring, we are able to model equivariance to higher-dimensional groups: we implement two $Sim(2)$-CNNs and evaluate them on MNIST-rot-scale. Lastly, to investigate the advantages equivariance brings in more complex problem settings, we experiment with all setups on three vision benchmark datasets; CIFAR10, CIFAR100 (Krizhevsky et al., 2009) and Galaxy10 (Leung & Bovy, 2019). See Appx. B.2 for more details on the datasets and training regimes used, and Appx. C.1 for an additional experiment empirically validating the equivariance properties of our models.

**Discretisation versus random sampling over compact subgroups** We investigate random sampling to approximate the group convolution. We trained $SE(2)$-CNNs on rotated MNIST for different resolutions over $SO(2)$, using random sampling and discretisation. Results are shown in Fig. 3. We can clearly see the advantage of approximating the group convolution integral through random sampling, likely attributable to the fact that through random sampling we obtain an unbiased estimator for the convolution, whereas discretisation equates to a biased sampling of the rotation group.

**Separable and non-separable G-CNN performance on MNIST variants** Following Cohen & Welling (2016a); Weiler et al. (2018b); Sosnovik et al. (2019); Finzi et al. (2020) we conduct experiments on MNIST variants with separable and non-separable implementations of equivariant models for each of the three groups. We discuss results per dataset in-depth.

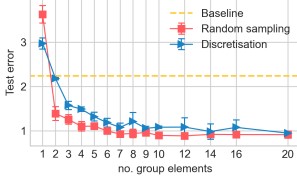 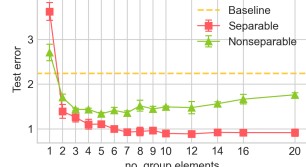 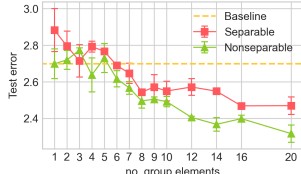

Figure 3: Test error vs. SO(2) resolution for separable SE(2)-CNNs on MNIST-rot, discretisation vs. random sampling.

Figure 4: Test error versus sampling resolution of SO(2) on MNIST-rot for separable and non-separable SE(2)-CNNs.

Figure 5: Test error versus sampled extent of the dilation group for separable and non-separable $\mathbb{R}^2 \rtimes \mathrm{R}^+$-CNNs.

***Rotated MNIST*** The SE(2)-CNNs are evaluated on rotated MNIST, the standard benchmark for rotation equivariant models. To assess the influence of sampling resolution over the group, we vary the number of sampled group elements from 1 to 20. The group convolution is approximated through random sampling. Results are shown in Fig. 4. We see the influence of increasing the resolution over the rotation group from 1 to 4 elements, after which the non-separable model saturates, and performance starts to drop when further increasing the number of sampled rotations. This is in line with findings by Bekkers et al. (2018), who note that as the group resolution increases, so does the possibility for overfitting on specific spatial configurations. The separable implementation saturates around 7 elements, but performance does not drop significantly beyond this point. This may imply that the reduction in kernel expressivity also has a regularising effect that benefits generalisation. In this experiment, separable group convolutions decisively outperform the non-separable variant.

***Scaled MNIST*** The $\mathbb{R}^2 \rtimes \mathbb{R}^+$-CNNs are evaluated on MNIST scale. The group convolution integral is approximated through discretisation. In this experiment, we retain the same resolution over the dilation group in all experiments, but instead vary the value at which the group is truncated, ranging from 1.0 at a grid of 1 element in the Lie algebra to 3.0 at a grid of 20 group elements. As in Sosnovik et al. (2019), we found that the inter-scale interactions in the group convolution operation reduced performance, likely because of increase in equivariance error due to truncation of the group. Therefore, we limited support of the group convolution kernel on $\mathbb{R}^+$ to two neighbouring elements for all experiments in this paper. Results are shown in Fig. 5. Here, non-separable group convolutions outperform the separable variant, suggesting that the ability of non-separable group convolutions to model spatial feature patterns over different scales is beneficial for the scaled MNIST dataset.

***Scaled rotated MNIST*** Lastly, the Sim(2)-CNN is benchmarked on MNIST-rot-scale. We limit the sampling resolution at each layer of the G-CNN to 2,4,6 and 8 elements for each subgroup. We test two implementations: (1) separable, where the group convolution is factorised into two convolutions, one over $\mathbb{R}^+ \rtimes \mathrm{SO}(2)$ and one over $\mathbb{R}^2$, and (2) $H$-separable, which factorises the convolution into three steps: a convolution over $\mathbb{R}^+$, one over $\mathrm{SO}(2)$, and one over $\mathbb{R}^2$. As shown in Fig. 6 the separable implementation turned out to be unstable for low SO(2) resolutions. We suspect that an approximation of SO(2) of only 2 rotation elements, paired with possible aliasing effects and equivariance error occurring over the dilation group, impedes the model from learning robust representations. For higher resolutions over SO(2), this effect seems to decrease. The results for the $H$-separable implementation (Fig. 7), highlight a much clearer increase in performance as the resolution over both groups increases, which suggests decoupling these information mappings stabilises the learning process.

**Application to computer vision benchmarks** To evaluate the value of equivariance to different transformations groups in natural image classifcation, we apply the SE(2)-, $\mathbb{R}^2 \rtimes \mathbb{R}^+$- and Sim(2)-CNNs to three vision datasets. For SE(2) and $\mathbb{R}^2 \rtimes \mathbb{R}^+$ we use resolutions of 4 and 8 group elements, and truncate the scale group at a value of $\sqrt{3}$, as we found this to work well for our experiments. For Sim(2), we use a resolution of 4 elements over the SO(2) group and 4 elements over the $\mathbb{R}^2 \rtimes \mathbb{R}^+$ subgroups and again truncate $\mathbb{R}^+$ at $\sqrt{3}$. On CIFAR10, as GPU memory allows, we also experiment with using resolutions of 8 group elements for both subgroups. As baseline, we implement the same architecture of our non-separable G-CNN, but restrict it to equivariance to translations. Results are summarised in Table 1 and exhibit the following general pattern: For most groups, separable group convolutions outperform non-separable ones, and in all experiments best performance is achieved by a Sim(2)-CNN. Next, we discuss results per dataset in-depth.

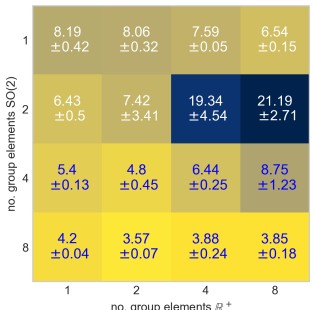
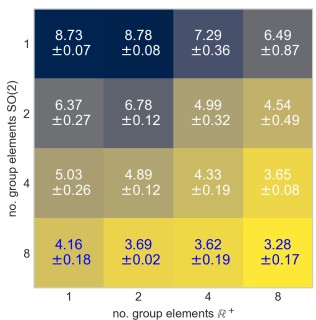
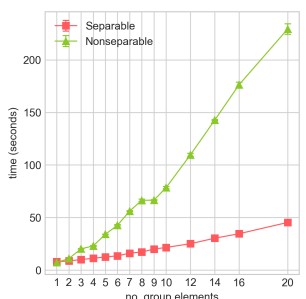

Figure 6: Test error versus resolution over rotation and scale group for separable $\mathrm{Sim}(2)-$CNN.

Figure 7: Test error versus resolution over rotation and scale group for H-separable $\mathrm{Sim}(2)-$CNN.

Figure 8: Process time per epoch (one pass over training and test sets) in seconds for different resolutions on $H$.

Table 1: Test accuracy (%) on different vision benchmark datasets. Best performance per group convolution implementation is underlined in blue. Best overall performance is boldfaced. † Separable along rotation and scale dimensions.

| GROUP | NO. ELEM | SEPARABLE | CIFAR10 | CIFAR100 | GALAXY10 |
|---|---|---|---|---|---|
| SE(2) | 4 | ✗ | 74.50(±0.14) | 48.39 (±1.28) | 85.58(±0.45) |
| | | ✓ | 78.30(±0.75) | 48.11 (±0.25) | 86.96(±0.07) |
| | 8 | ✗ | 76.81(±0.57) | 51.67 (±0.65) | 84.97(±0.07) |
| | | ✓ | 80.89(±0.85) | 50.19 (±0.40) | 86.61(±0.38) |
| $\mathbb{R} \rtimes \mathbb{R}^+$ | 4 | ✗ | 81.44(±0.62) | 44.80 (±0.56) | 85.13(±0.07) |
| | | ✓ | 83.08(±0.43) | 52.50 (±0.60) | 84.82(±0.25) |
| | 8 | ✗ | 80.56(±0.82) | 44.18(±0.92) | 84.83(±0.22) |
| | | ✓ | 83.60(±0.23) | 53.26 (±0.10) | 85.30(±0.06) |
| $\mathrm{Sim}(2)$ | 4 × 4 | ✓ | 86.63(±0.60) | 49.93 (±0.86) | 84.90(±0.19) |
| | | ✓† | 85.65(±0.66) | **54.62** (±1.41) | **87.45(±0.31)** |
| | 8 × 8 | ✓ | **89.38(±0.25)** | - | - |
| | | ✓† | 87.64(±0.16) | - | - |
| Baseline | - | - | 77.79(±0.27) | 47.91(±1.50) | 85.10(±0.42) |

***CIFAR10*** First, we evaluate our models on the CIFAR10 dataset (Krizhevsky et al., 2009). Results in Table 1 show that on CIFAR10, the non-separable variant of our SE(2)-CNN is outperformed by the translation equivariant baseline. With separable group convolutions and a group resolution of 8 elements, SE(2)-CNN show improved performance. Interestingly, $\mathbb{R}^2 \rtimes \mathbb{R}^+$-CNNs outperform both baseline and SE(2)-CNNs, so it seems scale invariance is a better inductive bias than rotation invariance in CIFAR10. Combining scale and rotation invariance ultimately leads to the best performance, with separable $\mathrm{Sim}(2)$-CNNs and a resolution of 8 scale and 8 rotation group elements achieving 89.38% test accuracy. The $H$-separable $\mathrm{Sim}(2)$-CNN, which models no interactions between scale and rotation, performs slightly worse.

***CIFAR100*** To assess the impact of separating spatial and group information in more complex problem settings, we experiment with CIFAR100. Notably, the non-separable $\mathbb{R}^2 \rtimes R^+$-CNNs perform well under baseline. We discuss this particular finding in Sec. 6. For SE(2)-CNNs, non-separable convolutions outperform separable ones, suggesting the model benefits from modelling distinct spatial configurations at different poses. Again, best performance is achieved by the $\mathrm{Sim}(2)$-CNN.

***Galaxy10*** Lastly, we apply our models in the domain of astro-photography, on the Galaxy10 dataset Leung & Bovy (2019). Rotation and scaling symmetries exist naturally in this dataset, making it an interesting application for our invariant models. Here, results show little difference between performance of non-separable and separable approaches. Rotation invariance seems to be the stronger inductive bias in this dataset, but again, combining rotation and scale invariance leads to the best performance, with the $H$-separable $\mathrm{Sim}(2)$-CNN achieving 87.45% test accuracy. The higher separable $\mathrm{Sim}(2)$ test error seems to indicate that interactions between scale and rotation features for this dataset impede model performance, again, possibly due to overfitting.

Table 2: Test error (%) on rotated MNIST for separable G-CNNs in comparison to other equivariant baselines: G-CNN (Cohen & Welling, 2016a), H-Net (Worrall et al., 2017), RED-NN (Salas et al., 2019), LieConv (Finzi et al., 2020), SFCNN (Weiler et al., 2018b), E(2)-NN (Weiler & Cesa, 2019). † Separable along dilation and rotations dimensions. + Train-time augmentation by continuous rotations.

| Baseline Methods | | | | | | | Separable G-CNNs (Ours) | | | |
|---|---|---|---|---|---|---|---|---|---|---|
| G-CNN | H-Net | RED-NN | LieConv | SFCNN | SFCNN$_+$ | E(2)-NN$_+$ | SE(2) | SE(2)$_+$ | Sim(2)$^\dagger$ | Sim(2)$^\dagger_+$ |
| 2.28 | 1.69 | 1.39 | 1.24 | 0.88 | 0.714 | 0.68 | $0.89_{\pm.008}$ | $0.66_{\pm.023}$ | $0.66_{\pm.009}$ | $\mathbf{0.59}_{\pm.008}$ |

Table 3: Test error (%) on CIFAR10 with All-CNN-C architecture, for our separable group convolutions in comparison to other baselines: All-CNN-C (Springenberg et al., 2014), $p4$- and $p4m$-G-CNN (Cohen & Welling, 2016a). † Separable along dilation and rotations dimensions. + Train-time augmentation by random horizontal flips and random cropping. $n$-Sim(2)-CNNs where $n$ is the SIREN hidden size in units. 6-Sim(2)-CNNs have approximately equal numbers of parameters to the original All-CNN-C.

| Baseline Methods | | | | | Separable G-CNNs (Ours) | | | | | |
|---|---|---|---|---|---|---|---|---|---|---|
| 1.4m param. | 1.37m param. | | 1.27m param. | | 1.14m param. | | 1.33m param. | | 3.22m param. | |
| All-CNN-C | $p4$-G-CNN | $p4$-G-CNN$_+$ | $p4m$-G-CNN | $p4m$-G-CNN$_+$ | 5-Sim(2)$^\dagger$ | 5-Sim(2)$^\dagger_+$ | 6-Sim(2)$^\dagger$ | 6-Sim(2)$^\dagger_+$ | 16-Sim(2)$^\dagger$ | 16-Sim(2)$^\dagger_+$ |
| 9.08 | 8.84 | 7.67 | 7.59 | 7.04 | 8.50 | 7.41 | 8.22 | 6.47 | 7.27 | **5.50** |

**Comparison in training time between separable and non-separable group convolutions** As one of the main motivators for the separable group convolution is its theoretical increase in computational efficiency, a comparison is made in training time between separable and non-separable G-CNNs. We tracked processing times per epoch, which consists of a train- and inference procedure, for the rotated MNIST experiments, and show them in Fig. 8. Separable group convolutions present a remarkable decrease in inference time, which may be leveraged to allow for model equivariance to larger groups.

**SOTA on rotated MNIST** To compare separable G-CNNs with related work, we finetune our models to competitive performance. With minimal adjustments to model and training regime, listed in App. B.2, we are able to achieve state of the art performance on MNIST-rot, see Tab. 2.

**Competitive performance on CIFAR10** We compare performance of our approach in a larger model. To this end, like (Cohen & Welling, 2016a), we re-implement the All-CNN-C architecture by Springenberg et al. (2014), using our Sim(2) convolution layers as drop-in replacement. To also compare performance in absolute numbers of trainable parameters, we train three configurations, see App. B.2. We show that Sim(2)-CNNs, with SIREN hidden sizes of 6 and 5 units, containing similar or smaller numbers of trainable parameters to the original All-CNN-C, improve model accuracy compared to CNNs or G-CNNs equivariant to Sim(2)-subgroups $\mathbb{R}^2$ and $p4$, and with data augmentation 6-Sim(2)-CNNs also outperform $p4m$-G-CNNs, see Tab. 3. Increasing SIREN hidden size to 16 leads to even more significant improvements in performance.

## 6 DISCUSSION & FUTURE WORK

**On implicit kernel representations** Some of the non-separable configurations generated results below baseline, which is unexpected given that G-CNNs usually improve upon regular CNNs. We conjecture that these results are caused by the SIREN parameterisation, which take as input a coordinate vector of different units (mixing both spatial and sub-group $h$ coordinates). This could limit kernel expressivity, as sharing of a frequency multiplier $\omega_0$ sets kernel smoothness to be identical along spatial and subgroup dimensions. The fact that for the separable variant we use two distinct SIRENs would lift this restriction. Despite useful advantages of MLP parametrisations for the kernels, such as their flexibility and ease of implementation for arbitrary Lie groups, it remains an open problem to have full control over their smoothness; ideally one band-limits the kernel MLPs to the discretised resolution on each axis, which would be an important direction for future work.

**Conclusion** Motivated by observed redundancies in learned group convolution filters, we introduced separable group convolutions, a computationally efficient implementation of regular group convolutions. We showed that separable group convolutions not only drastically increase computational efficiency, but in many settings also outperform their non-separable counterpart. Furthermore, we demonstrated the value of separable group convolutions as a solution for modelling equivariance to larger groups; Sim(2)-CNNs decisively outperform models only equivariant to subgroups of Sim(2), clearly reinforcing equivariance as means of model generalisation.

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

## A  THEORY AND BACKGROUND

### A.1  GROUP THEORETIC PRELIMINARIES

In this section, group theoretical prerequisites used throughout the paper are briefly refreshed. This is by no means intended as an exhaustive exposition of the fundamentals of group theory, we only introduce those concepts relevant to the current work.

**Group.** A group is defined by a set $G$ of *group elements*, along with a binary operator $\cdot : G \times G \to G$, called the *group product*. The group product defines a way to combine each pair of elements $g_1, g_2 \in G$. For the binary operator $\cdot$ to be considered a group product, it needs to satisfy four constraints:

1. Closure. $G$ is closed under $\cdot$; for all $g_1, g_2 \in G$ we have $g_1 \cdot g_2 \in G$.
2. Identity. There exists an identity element $e$ s.t. for each $g \in G$, we have $e \cdot g = g \cdot e = g$.
3. Inverse. For every element $g \in G$ we have an element $g^{-1} \in G$, s.t. $g \cdot g^{-1} = e$.
4. Associativity. For every set of elements $g_1, g_2, g_3 \in G$, we have $(g_1 \cdot g_2) \cdot g_3 = g_1 \cdot (g_2 \cdot g_3)$.

**Lie groups.** A Lie group is a group of which the elements form a smooth manifold. Since the group itself is not necessarily a vector space, combination of elements through addition or subtraction is not defined. However, to each Lie group $G$, an algebra $\mathfrak{g}$ may be associated, given by the tangent space of the Lie group at the identity $T_e(G)$. The Lie algebra may be interpreted as a vector space of infinitesimal generators of the group, a set of elements from which we can obtain the group $G$, by repeated application.

**Exponential and logarithmic map.** The exponential map $\exp : \mathfrak{g} \to G$ is a function mapping elements from the Lie algebra to the group. For many transformation Lie groups of interest this map is surjective, and it is possible to define an inverse mapping; the logarithmic map which maps from the group to the algebra.

**Semi-direct product groups.** In practice, we are often only interested in data defined over $\mathbb{R}^d$, and hence in this paper only consider affine Lie groups of the form $G = \mathbb{R}^d \rtimes H$, where $\mathbb{R}^d$ is the translation group in $d$ dimensions and $H$ is a transformation Lie group of interest.

**Group action.** A group $G$ may have an action on a given space $\mathcal{X}$. Given a group element $g \in G$ and a set $\mathcal{X}$, the group action $\mathcal{T}_g$ defines what happens to any element of $x \in \mathcal{X}$ when we apply the transformation given by element $g$ to it. This action is given by:

$$\mathcal{T} : G \times \mathcal{X} \to \mathcal{X} \text{ and } \mathcal{T}_g : \mathcal{X} \to \mathcal{X}, \tag{6}$$

such that for any two elements $g, h \in G$, we can combine their actions into a single action; $\mathcal{T}_{g,h} = \mathcal{T}_g \circ \mathcal{T}_h$. To avoid clutter, we write the action $\mathcal{T}_g(x)$ as $g \cdot x$. Note that the action of a group $G$ on domain $\mathcal{X}$ also extends to functions defined on this domain, treated next.

**Left-regular representations** We extend the group action on $\mathcal{X}$ to *square integrable functions* defined on $\mathcal{X}$; $\mathbb{L}_2(X)$. Intuitively, any group action on $\mathcal{X}$ induces an action on functions on $\mathcal{X}$; as elements of the set $\mathcal{X}$ are transformed, a function on $\mathcal{X}$ is *dragged along*. Commonly, this is

expressed through the *left-regular representations*. Imagine we have a function; $f : \mathcal{X} \to \mathbb{R}$. Let's say we want to reason about the function $f$ after transformation by group element $r$; let us denote this transformed function $f'$. We may inspect the value of this function for a given element of $\mathcal{X}$ by reasoning backwards from our transformed function. For example, to obtain the value of $f'$ for the transformed element $a'$, we find what the value of $f$ was for $a$ before applying $r$. This is done by applying the inverse of action $r$ to the transformed element $a'$. For any element of the set of square-integrable functions on $\mathcal{X}$; $f \in \mathbb{L}_2(\mathcal{X})$ the left-regular representation of $g$ is given by:

$$\mathcal{L}_g : f \to f' \text{ and for } a' \in \mathcal{T}_g(\mathcal{X}): f'(a') = f(\mathcal{T}_{g^{-1}}(a')). \tag{7}$$

**Equivariance**. An operator is equivariant with respect to a group, if it commutes with the action of the group. For an operator $\Phi : \mathbb{L}_2(X) \to \mathbb{L}_2(Y)$:

$$\forall g \in G : \mathcal{L}_g \circ \Phi = \Phi \circ \mathcal{L}_g. \tag{8}$$

## A.2 Deriving separable group convolutions

If we set the kernel $k$ to be separable, meaning we parameterise it by multiplying a kernel $k_H$ which is constant along the spatial domain and a kernel $k_{\mathbb{R}^2}$ which is constant along the group domain:

$$\begin{aligned} k(g) &= k(\boldsymbol{x}, h) \\ &= k_{\mathbb{R}^2}(\boldsymbol{x}) k_H(h). \end{aligned} \tag{9}$$

We can derive separable group convolutions as:

$$\begin{aligned} (f *_{\text{group}} k)(g) &= \int_G f(\tilde{g}) k(g^{-1} \cdot \tilde{g}) \, \mathrm{d}\mu(\tilde{g}) \\ &= \int_{\mathbb{R}^2} \int_H f(\tilde{\boldsymbol{x}}, \tilde{h}) \mathcal{L}_{x^{-1}} \mathcal{L}_{h^{-1}} k(\tilde{\boldsymbol{x}}, \tilde{h}) \frac{1}{|h|} \, \mathrm{d}\tilde{\boldsymbol{x}} \, \mathrm{d}\tilde{h} \\ &= \int_{\mathbb{R}^2} \int_H f(\tilde{\boldsymbol{x}}, \tilde{h}) k(h^{-1}(\tilde{\boldsymbol{x}} - \boldsymbol{x}), h^{-1} \cdot \tilde{h}) \frac{1}{|h|} \, \mathrm{d}\tilde{\boldsymbol{x}} \, \mathrm{d}\tilde{h} \\ &\to \int_{\mathbb{R}^2} \int_H f(\tilde{\boldsymbol{x}}, \tilde{h}) k_{\mathbb{R}^2}(h^{-1}(\tilde{\boldsymbol{x}} - \boldsymbol{x})) k_H(h^{-1} \cdot \tilde{h}) \frac{1}{|h|} \, \mathrm{d}\tilde{\boldsymbol{x}} \, \mathrm{d}\tilde{h} \\ &= \int_{\mathbb{R}^2} \int_H f(\tilde{\boldsymbol{x}}, \tilde{h}) k_H(h^{-1} \cdot \tilde{h}) k_{\mathbb{R}^2}(h^{-1}(\tilde{\boldsymbol{x}} - \boldsymbol{x})) \frac{1}{|h|} \, \mathrm{d}\tilde{\boldsymbol{x}} \, \mathrm{d}\tilde{h} \\ &= \int_{\mathbb{R}^2} \left[ \int_H f(\tilde{\boldsymbol{x}}, \tilde{h}) k_H(h^{-1} \cdot \tilde{h}) \, \mathrm{d}\tilde{h} \right] k_{\mathbb{R}^2}(h^{-1}(\tilde{\boldsymbol{x}} - \boldsymbol{x})) \frac{1}{|h|} \, \mathrm{d}\tilde{\boldsymbol{x}}. \end{aligned} \tag{10}$$

## A.3 Examples of different Lie groups

We implemented models equivariant to three different groups: $\mathrm{SE}(2)$, $\mathbb{R}^2 \rtimes \mathbb{R}^+$ and $\mathrm{Sim}(2)$. In this section, we describe these groups in more detail, and give definitions for the logarithmic map required in obtaining G-CNNs equivariant to these groups (Bekkers, 2019).

**The translation group** $\mathbb{R}^2$ The translation group in two dimensions $\mathbb{R}^2$, has group product and inverse for two elements $g = \boldsymbol{x}, g' = \boldsymbol{x}' \in \mathbb{R}^2$:

$$g \cdot g' = (\boldsymbol{x} + \boldsymbol{x}') \tag{11}$$
$$g^{-1} = -\boldsymbol{x}. \tag{12}$$

With logarithmic map:

$$\log g = \boldsymbol{x}. \tag{13}$$

**The rotation group** $\mathrm{SO}(2)$ The rotation group in two dimensions describes the set of continuous rotation transformations of the plane, and consists of all orthogonal matrices $\boldsymbol{R}$ with determinant 1. Its group product and inverse for two elements $g = \boldsymbol{R}_\theta, g' = \boldsymbol{R}'_\theta \in \mathrm{SO}(2)$ is given by:

$$\begin{aligned} g \cdot g' &= \boldsymbol{R}_\theta \boldsymbol{R}_{\theta'} \\ &= \boldsymbol{R}_{\theta+\theta'} \end{aligned} \tag{14}$$
$$g^{-1} + \boldsymbol{R}_\theta^{-1}. \tag{15}$$

With logarithmic map:

$$\log g = \begin{bmatrix} 0 & -\theta \mod 2\pi \\ \theta \mod 2\pi & 0 \end{bmatrix}. \tag{16}$$

**The dilation group** $\mathbb{R}^+$ The group of dilation transformations $\mathbb{R}^+$ has a group product and inverse that, for two elements $g = s, g = s' \in \mathbb{R}^+$ are given by:

$$g \cdot g' = ss' \tag{17}$$

$$h^{-1} = s^{-1}. \tag{18}$$

The logarithmic map is given by:

$$\log g = \ln s. \tag{19}$$

**The Special Euclidean group SE**$(2)$ The Special Euclidean group in 2 dimensions describes the set of geometric transformations that are formed by combinations of rotations and translations in two dimensions. Each group element can be parameterised by two variables $\theta$ and $\mathbf{x}$, describing the rotation angle and translation vector, $g = (\theta, \mathbf{x}) \in G$. For two elements $g, g' \in \mathrm{SE}(2)$, the group product and inverse are given by:

$$g \cdot g' = (\mathbf{x}, \theta) \cdot (\mathbf{x}', \theta')$$
$$= (\mathcal{T}_\theta(\mathbf{x}') + \mathbf{x}, \theta + \theta') \tag{20}$$
$$g^{-1} = (-\mathcal{T}_{-\theta}(\mathbf{x}), -\theta). \tag{21}$$

As we can see, to combine the two elements $g, g'$ we apply the action of the rotation part of $g$ to the translation of $g'$ before we combine it with the translation of $g$, we combine elements semi-directly. $\mathrm{SE}(2)$ is a semidirect product of the translation group $\mathbb{R}^2$ and rotation group $\mathrm{SO}(2)$. We write this as $\mathrm{SE}(2) = \mathbb{R}^2 \rtimes \mathrm{SO}(2)$. To simplify implementation, we separate the logarithmic map into logarithmic maps for $\mathrm{SO}(2)$ and $\mathbb{R}^2$ in our implementation.

**The dilation-translation group** $\mathbb{R}^2 \rtimes \mathbb{R}^+$ Another group of interest to our research topic is the translation-dilation group $\mathbb{R}^2 \rtimes \mathbb{R}^+$, the group of translation and dilation transformations. Dilations occur frequently in natural images, in the form of scaling transformations of objects and scenes, as the distance between camera and object differs between images. Group elements are parameterised by a scaling factor $s$ and translation element $\mathbf{x}$, $g = (\mathbf{x}, s) \in G$. For two elements $g, g'$, group product and inverse are given by:

$$g \cdot g' = (\mathbf{x}, s) \cdot (\mathbf{x}', s')$$
$$= (\mathcal{T}_s(\mathbf{x}') + \mathbf{x}, ss') \tag{22}$$
$$g^{-1} = (-\mathcal{T}_{s^{-1}}(\mathbf{x}), s^{-1}). \tag{23}$$

To simplify implementation, we separate the logarithmic map into logarithmic maps for $\mathbb{R}^+$ and $\mathbb{R}^2$ in our implementation.

**The Similarity group Sim**$(2)$ The similarity transformation group is the semi-direct product of the roto-translation group $\mathrm{SE}(2)$ and the isotropic scaling group $\mathbb{R}^+$, and defines dilation-roto-translation transformations in two dimensions. Each group element can be parameterised by three variables $\theta$, $s$ and $\mathbf{x}$. For two elements $g, g' \in \mathrm{Sim}(2)$ the group product and inverse are given by:

$$g \cdot g' = ((\mathbf{x}, \theta), s) \cdot ((\mathbf{x}', \theta'), s')$$
$$= ((\mathcal{T}_s(\mathcal{T}_\theta(\mathbf{x}')) + \mathbf{x}, \mathcal{T}_s(\theta') + \theta), ss')$$
$$= ((\mathcal{T}_s(\mathcal{T}_{\mathcal{T}_s(\theta)}(\mathbf{x}')) + \mathbf{x}, \mathcal{T}_s(\theta') + \theta), ss')$$
$$= ((\mathcal{T}_s(\mathcal{T}_\theta(\mathbf{x}')) + \mathbf{x}, \theta' + \theta), ss') \tag{24}$$
$$g^{-1} = (-(\mathcal{T}_{s^{-1}}(\mathcal{T}_{-\theta}(\mathbf{x})), -\theta), s^{-1}). \tag{25}$$

In Eq. 24 we use the fact that the isotropic dilation group has no action on the rotation group. This is a consequence of the fact that $\mathbb{R}^+$ and $\mathrm{SO}(2)$ are both abelian groups, and may be taken in direct product to create the group of dilation-rotation transformations. We again separate logarithmic maps by subgroup.

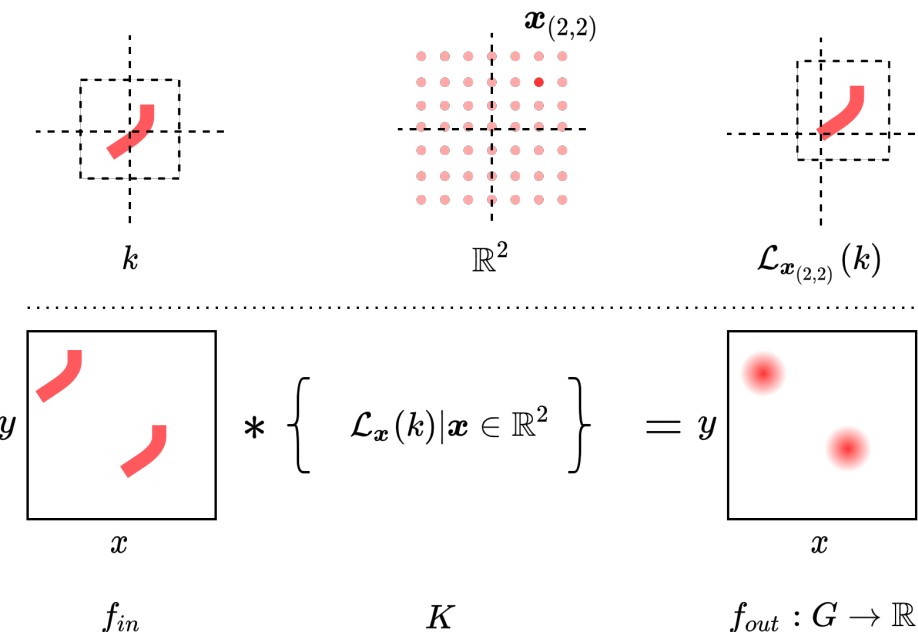

Figure 9: The convolution operation in a CNN. In upper section of the figure above, a spatial kernel $k$ is transformed by group element $\boldsymbol{x}_{(2,2)} \in \mathbb{R}^2$ to yield $\mathcal{L}_{\boldsymbol{x}_{(2,2)}}(k)$. In the lower section, $k$ is transformed under the action of each element of (a discretisation of) the group $\mathbb{R}^2$, to yield a set $K$ of translated copies of $k$. Convolving over $f_{in}$ with each of these copies yields a new feature map $f_{out}$ defined over $G = \mathbb{R}^2$.

## A.4 VISUALISATIONS OF OPERATIONS IN (SEPARABLE) G-CNNS

To ease the reading experience of this work, in this section we provide some additional group theoretic perspective on regular CNNs, and additional visualisations and for a number of operations used in (separable) group convolutions.

**CNNs from a group theoretic perspective** We give a brief treatment of ordinary CNNs The ordinary convolution operation used in neural networks requires a definition of a convolution kernel $k$ on $\mathbb{R}^2$, as we are modulating a signal $f$ which itself lives on $\mathbb{R}^2$. The kernel $k$ is applied to $f$ on every location in the input space $\mathbb{R}^2$ to again yield a function over $\mathbb{R}^2$. Intuitively, this is the same as saying (1) we transform the convolution kernel $k$ under the action of every group element $\boldsymbol{x} \in \mathbb{R}^2$, to obtain a set of kernels $K = \{\mathcal{L}_{\boldsymbol{x}}(k)|\boldsymbol{x} \in \mathbb{R}^2\}$, and (2) apply the convolution operation on $f$ using this set of transformed kernels $K$. By *tying* the kernel weights used throughout the translation group, learned features are automatically generalised over spatial positions. This intuition is visualised in Fig. 9.

**Lifting convolution** In Fig. 10 we show a visualisation of the lifting convolution for the SE(2) group.

**Group convolution** In Fig. 11 we show a visualisation of the group convolution for the SE(2) group.

**Separable group convolution kernel** In Fig. 12 we show a visualisation of a separable group convolution kernel.

**Separable group convolution** In Fig. 13 we show a visualisation of the separable group convolution operation.

**Defining convolution kernels on Lie groups** In Fig. 15 we show how we obtain a grid on the Lie group SO(3)/SO(2) by mapping from its algebra. In Fig. 14 we show how we subsequently obtain a kernel on this group by defining a SIREN on its algebra.

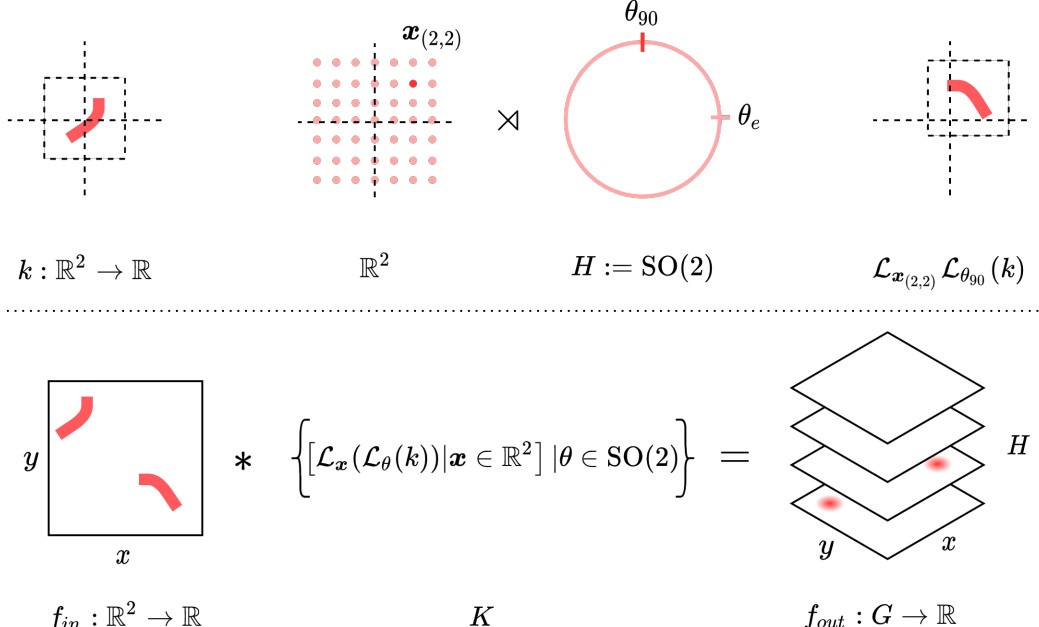

Figure 10: An example of the lifting convolution for $G = \mathrm{SE}(2) = \mathbb{R}^2 \rtimes \mathrm{SO}(2)$. In the upper section of the above figure, a spatial kernel $k$ is transformed by group element $(\boldsymbol{x}, \theta) \in \mathrm{SE}(2)$ with $\boldsymbol{x} = (2, 2), \theta = 90°$ to yield $\mathcal{L}_{\boldsymbol{x}_{(2,2)}} \mathcal{L}_{\theta_{90}}(k)$. In the lower section, $k$ is transformed under the action of each element of (a discretisation of) the group $\mathrm{SE}(2)$, to yield a set $K$ of translated and rotated copies of $k$. Convolving $f_{in}$ with each of these copies yields a new feature map $f_{out}$ defined over $G = \mathrm{SE}(2)$.

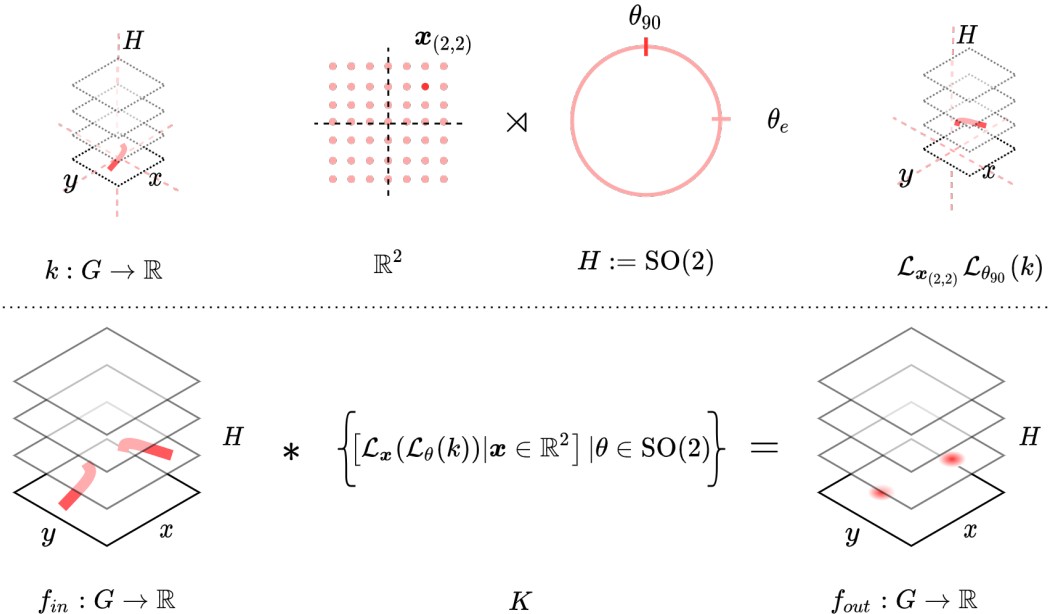

Figure 11: An example of the group convolution for (a discretisation of) $G = \text{SE}(2) = \mathbb{R}^2 \rtimes \text{SO}(2)$. In the upper section of the above figure, a group convolution kernel $k$ (defined over $G$) is transformed by group element $(\boldsymbol{x}, \theta) \in \text{SE}(2)$ with $\boldsymbol{x} = (2,2), \theta = 90°$ to yield $\mathcal{L}_{\boldsymbol{x}_{(2,2)}} \mathcal{L}_{\theta_{90}}(k)$. In the lower section, $k$ is transformed under the action of each element of (a discretisation of) the group $\text{SE}(2)$, to yield a set $K$ of translated and rotated copies of $k$. Convolving $f_{in}$ with each of these copies yields a new feature map $f_{out}$ defined over $G = \text{SE}(2)$.

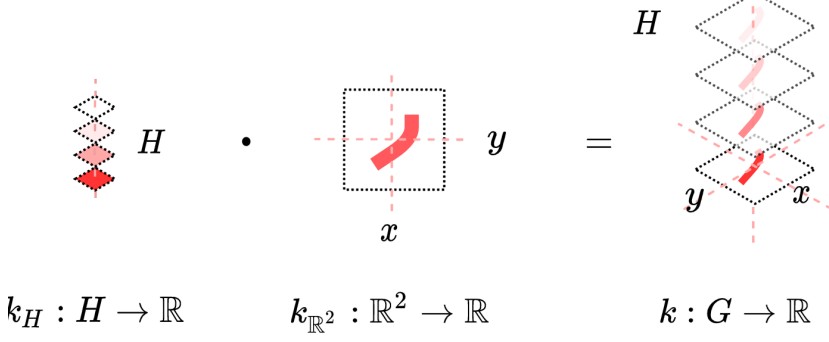

Figure 12: The separable group convolution kernel parameterises the kernel $k : G \rightarrow \mathbb{R}$ as $k_H k_{\mathbb{R}^2}$. $k_H$ is defined over the subgroup $H$, with no spatial extent, whereas $k_{\mathbb{R}^2}$ is defined only on $\mathbb{R}^2$ with no extent over the group. The full group convolution kernel $k$ is obtained by repeating $k_{\mathbb{R}^2}$ along $H$, weighting each instance of $k_{\mathbb{R}^2}$ by its corresponding value for $h \in k_H$. In practice, it is computationally advantageous to apply the convolution operations in sequence.

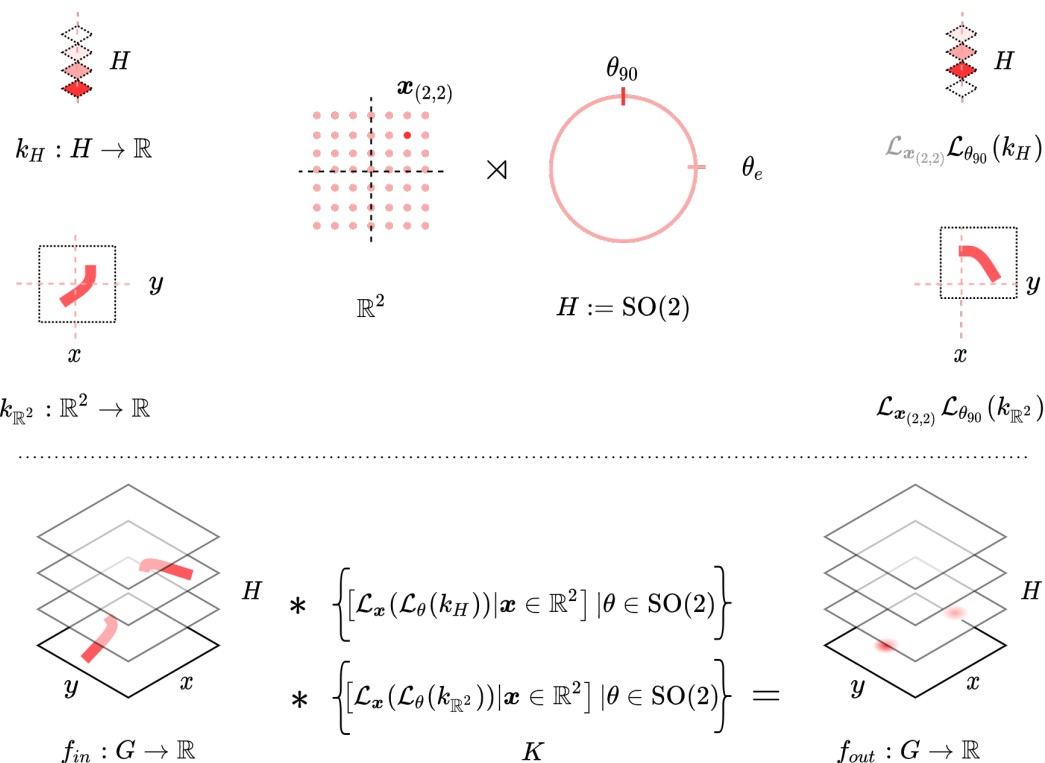

Figure 13: An example of the separable group convolution for (a discretisation of) $G = \mathrm{SE}(2) = \mathbb{R}^2 \rtimes \mathrm{SO}(2)$. In the upper section of the above figure, a kernel $k_H$ (defined over $H$) and a kernel $k_{\mathbb{R}^2}$ (defined over $\mathbb{R}^2$) are transformed by group element $(\boldsymbol{x}, \theta) \in \mathrm{SE}(2)$ with $\boldsymbol{x} = (2, 2), \theta = 90°$ to yield $\mathcal{L}_{\boldsymbol{x}_{(2,2)}} \mathcal{L}_{\theta_{90}}(k_H)$ and $\mathcal{L}_{\boldsymbol{x}_{(2,2)}} \mathcal{L}_{\theta_{90}}(k_{\mathbb{R}^2})$. In the lower section, $k_H$ and $k_{\mathbb{R}^2}$ are transformed under the action of each element of (a discretisation of) the group $\mathrm{SE}(2)$, to yield sets $K_H$ of transformed copies of $k_H$ and a set $k_{\mathbb{R}^2}$ of translated and rotated copies of $k$. Convolving $f_{in}$ with $K_H$ and $K_{\mathbb{R}^2}$ sequentially yields a feature map $f_{out}$ defined over $G = \mathrm{SE}(2)$.

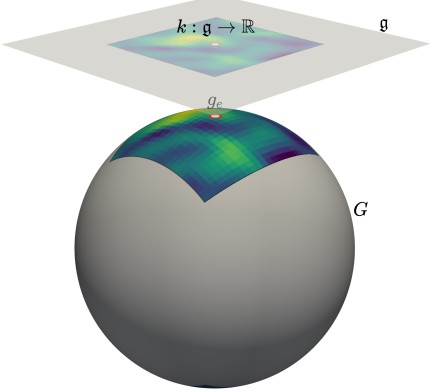

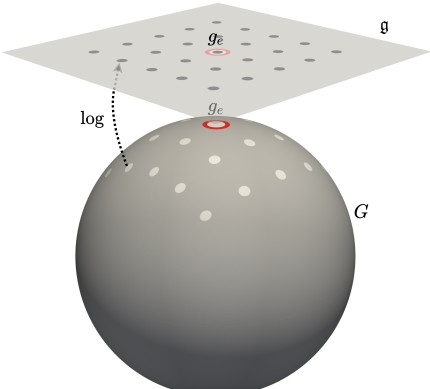

Figure 14: An example of a localised kernel $k$ on the quotient group $\mathrm{SO}(3)/\mathrm{SO}(2)$. Although the SIREN parameterising the kernel is defined on $\mathfrak{g}$, we can associate the kernel values sampled on $\mathfrak{g}$ with the relevant elements $g' \in G$ through use of the grid $\mathcal{H}$ defined on $G$, which we map to $\mathfrak{g}$ via the logarithmic map.

Figure 15: An example of a local kernel grid $\mathcal{H}$ on the quotient group $\mathrm{SO}(3)/\mathrm{SO}(2)$. We obtain a volumetrically constant grid on $G$ by sampling a set of equidistant points in $\mathfrak{g}$ and mapping them to $G$ via the exponential map. The grid on $G$, show in this figure, then serves as input for the SIREN defined on $\mathfrak{g}$ via the logarithmic map.

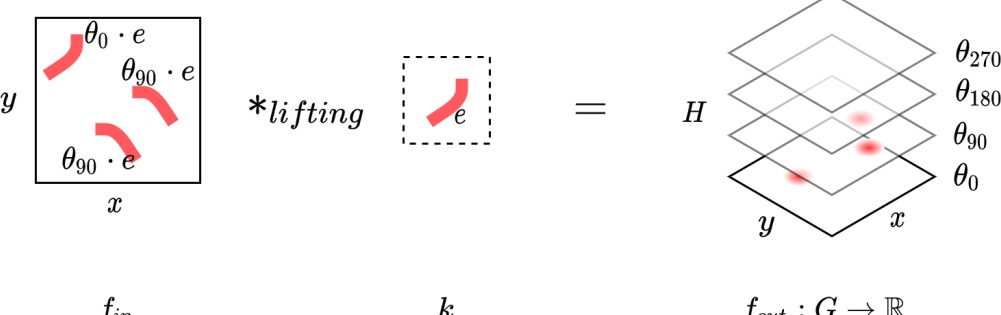

Figure 16: An example of a configuration of features $f_{out}$ a *single* separable group convolution kernel is unable to represent.

## A.5 Expressivity Limitations of Separable Group Convolutions

As discussed, separable group convolution kernels are strictly less expressive than their non-separable counterpart. For example, in the case of the roto-translation group $\mathrm{SE}(2)$, this would enable non-separable group convolution kernels to learn to represent features that are built up of different spatial configurations at different orientations.

We draw up a simplified example illustrated in Fig. 16. Assume we have an elementary feature type $e$, and a spatial kernel $k$ with which we can recognise this feature type in its canonical pose $\theta_0$. In our input $f_{in}$, we have three instances of the feature $e$, one under the canonical pose $\theta_0$, and two under a 90° rotation. Applying the lifting convolution using kernel $k$ for the group $G = \mathbb{R}^2 \rtimes C_4$ of translations and 90° rotations yields a feature map $f_{out}$ defined over $G$, with spatial feature maps for $\theta_0, ..., \theta_{270}$. The spatial feature map $f_{out}^{\theta_0}$ contains a response at a single spatial position. In contrast, the feature map $f_{out}^{\theta_{90}}$ contains a response at two spatial positions. The spatial configurations for the feature maps along $H$ are different. A single conventional group convolution kernel could learn to recognise these distinct spatial configurations along the subgroup axis, whereas a separable group

convolution kernel could not, since it simply repeats (a weighted version of) the same spatial kernel $k_{\mathbb{R}^2}$ along the group axis.

Although this reduction in expressivity could theoretically prove limiting in the application of G-CNNs on vision tasks, our experiments show that in practice this rarely seems a problem, and may even help prevent overfitting.

### A.6    KERNEL SMOOTHNESS FOR GROUP CONVOLUTIONS ON CONTINUOUS GROUPS

As mentioned, using SIRENs we are able to explicitly control kernel smoothness. We briefly elaborate on the importance of kernel smoothness in G-CNNs. In conventional CNNs, weights at distinct spatial locations are generally initialised independently. Because the kernels are only transformed using discrete translation operations $\boldsymbol{x} \in \mathbb{Z}^2$, translation equivariance is ensured by virtue of using the exact same weights values throughout all spatial locations.

In G-CNNs for continuous groups, the kernel is transformed under actions of a continuous transformation group of interest $H$ to obtain equivariance. However, in our convolution operation we are still using a discretised kernel; we are required to sample kernel values at *different grid points* for different elements $h \in H$. We are no longer able to simply reuse the same weight values throughout the group as with regular CNNs. To this end, we define our kernels in an analytical form, which we can trivially evaluate at arbitrary grid points. Because our grid has a fixed resolution, the kernels sampled from this analytical form are susceptible to aliasing effects; the analytical kernel function may exhibit higher frequencies than can be captured in the discretisation of the kernel. We visualise the effects of aliasing in Fig. 17.

In short, for continuous groups, the group action transforms a signal smoothly as the group is traversed. To prevent discretisation artefacts, we want our kernel to exhibit the same smoothly transforming behaviour, hence we use SIRENs, as they offer explicit control over the smoothness of kernels in their analytical form.

## B    ADDITIONAL EXPERIMENTAL DETAILS

### B.1    ARCHITECTURAL DETAILS AND PARAMETERISATION

**Model architecture** As architecture for our experiments we use a simple ResNet model (He et al., 2016). We use a single lifting convolution with 32 output channels, followed by two residual blocks of 2 group convolutions. The first block has 32 output channels, the second has 64 output channels. After the first residual block we apply max-pooling with kernel size 2 over the spatial dimensions of the feature map. After the last residual block, we apply max pooling over remaining spatial and subgroup dimensions, followed by two linear layers with batchnorm and ReLU in between. An overview is given in Fig. 18.

**Group convolution blocks and random sampling on the group** In our residual blocks (He et al., 2016), we subsequently convolve the input to the block $\boldsymbol{x}_{\text{in}}$ by two group convolutions, gconv_1 and gconv_2, yielding $\boldsymbol{x}_{\text{out}}$ and apply elementwise addition with the input $\boldsymbol{x}_{\text{in}}$, followed by ReLU activation.

When approximating the group convolution through random sampling, we must take care to define the input and output of the two group convolution layers on the same grid over the group as the skip-connection to ensure a well-defined equivariant group convolution block. This may be done by adding a group shortcut layer which maps from the set of input elements on the group to the set of output elements. We implement this as a group convolution with a $1 \times 1$ spatial extent. The group shortcut layer thus simultaneously serves as a channelwise projection from the input space of gconv_1 to the output space of gconv_2 *and* maps from the input grid on $H$ of the first group convolution to the output grid on $H$ of the second group convolution.

**SIREN architecture and separable kernel parameterisation** All our kernels are parameterised by a SIREN (Sitzmann et al., 2020). In a SIREN, output $\boldsymbol{y}^l$ for a layer $l$ and input $\boldsymbol{x}^{l-1}$ is defined by:

$$\boldsymbol{y}^l = \sin(\omega_0 \boldsymbol{W}^l \boldsymbol{x}^{l-1} + \boldsymbol{b}^l) \tag{26}$$

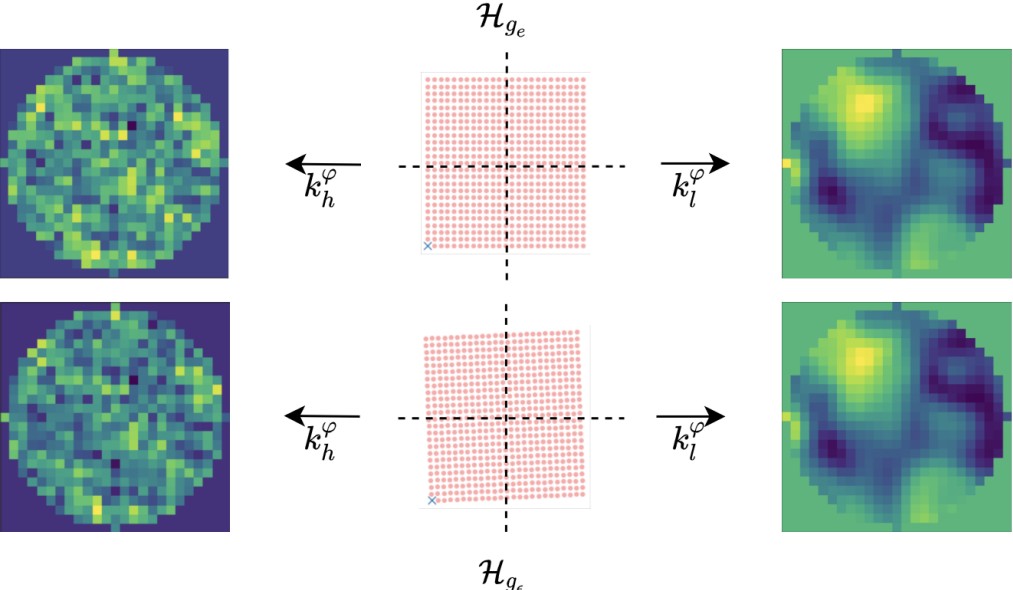

Figure 17: Example of aliasing effects in analytical kernel parameterisations. Assume we are performing a lifting convolution to the $SE(2)$ group. We evaluate two variants of our kernel function; $k_l^\varphi$ exhibiting low frequencies over the kernel domain, and $k_h^\varphi$ exhibiting high frequencies over kernel domain. We evaluate these functions for a spatial grid $\mathcal{H}_{g_e}$ to obtain a kernel under the identity rotation, and for a slightly rotated version of this grid $\mathcal{H}_{g_\epsilon}$. We see the kernel sampled from $k_l^\varphi$ changes smoothly, whereas the high frequency components in $k_h^\varphi$ lead to considerably different results for the two grids.

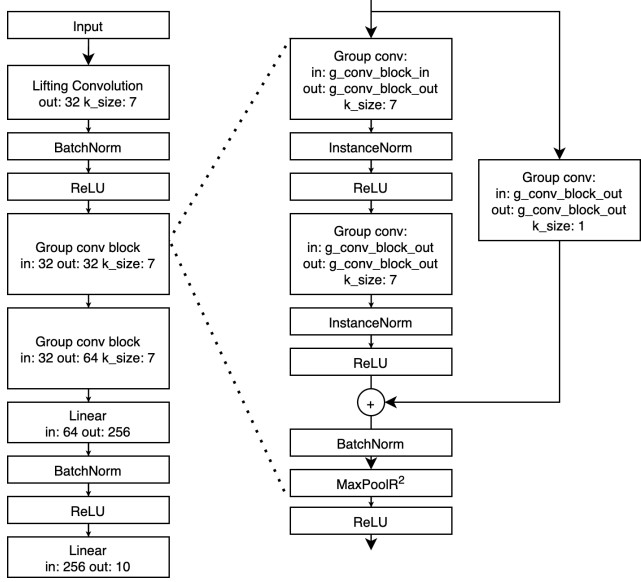

Figure 18: Architecture used over all experiments.

Table 4: Number of trainable parameters for different implementations. For all groups and datasets, these numbers are kept constant.

| NON-SEPARABLE | SEPARABLE | H-SEPARABLE |
|---|---|---|
| 742k | 803k | 864k |

In this equation, $\omega_0$ acts as a multiplier for low dimensional frequencies found in the input domain (the grid of relative offsets on the group), which explicitly introduces higher frequencies, allowing the neural net to learn high-frequency functions (such as kernels). We found a value for $\omega_0$ of 10 to work well in all our experiments. For the SIREN, we used an architecture of two hidden layers of 64 units. In non-separable G-CNNs we have a single SIREN with a final layer mapping to a vector $\mathbb{R}^{c_{in} \times c_{out}}$. In separable G-CNNs, we use two SIRENs, the first mapping to a function the Lie algebra of the subgroup $H$; $\mathbb{R}^{c_{in} \times c_{out}}$, and the second mapping to $\mathbb{R}^{c_{out}}$. Formulating the kernel $k$ for a group element $g = (\boldsymbol{x}, h)$ in terms of $k_H$, $k_{\mathbb{R}^2}$, input channel $i$ and output channel $j$, and logarithmic map on H $\log_H$, we obtain:

$$k^{i,j}(g) = k_H^{i,j}(\log_H h) k_{\mathbb{R}^2}^j(\boldsymbol{x}) \tag{27}$$

Lastly, for H-separable $\text{Sim}(2)$-CNNs we use three SIRENs, the first mapping to a function the Lie algebra of the subgroup $\mathbb{R}^+$; $\mathbb{R}^{c_{in} \times c_{out}}$, the second mapping $\text{SO}(2)$ to $\mathbb{R}^{c_{out}}$, and the third mapping $\mathbb{R}^2$ to $\mathbb{R}^{c_{out}}$. one mapping to $\text{SO}(2)$. Formulating the kernel $k$ for a group element $g = ((\boldsymbol{x}, \theta), s)$ in terms of $k_{\text{SO}(2)}$, $k_{\mathbb{R}^+}$, $k_{\mathbb{R}^2}$, input channel $i$ and output channel $j$, logarithmic maps $\log_{\mathbb{R}^+}$ and $\log_{\text{SO}(2)}$ on $\mathbb{R}^+$ and $\text{SO}(2)$ respectively we obtain:

$$k^{i,j}(g) = k_{\mathbb{R}^+}^{i,j}(\log_{\mathbb{R}^+} s) k_{\text{SO}(2)}^j(\log_{\text{SO}(2)} \theta) k_{\mathbb{R}^2}^j(\boldsymbol{x}) \tag{28}$$

**Model sizes** In Tab. 4, we report the number of trainable parameters for each model configuration. Throughout our experiments, we kept the number of channels in all our configurations constant, to fairly compare the expressivity of the learned representations in non-separable and separable group convolutions. This has as effect that the number of parameters in the separable implementations is larger than for non-separable implementations, due to our use of separate SIRENs to parameterise kernels over the different subgroups. To ensure that the difference in number of trainable parameters does not influence the comparison between separable and non-separable group convolution layers, we explicitly chose to over-parameterise our SIREN architecture, as shown in an additional ablation in App. C.3. We only change SIREN hidden size when comparing our models to baselines proposed in related works as detailed in App. B.2.

## B.2 EXPERIMENTAL DETAILS

Here, we list training regimes for all experiments. We keep these as consistent as possible, as we are only interested in the effect of separating the group convolution operation, and in isolating the effect of incorporating equivariance into our models. We list information on all datasets, and any dataset-specific model configuration details we use in our experiments here.

**Optimizer** All architectures are trained with Adam optimisation (Kingma & Ba, 2014), and $1e^{-4}$ weight decay. All models are trained on a single Titan V.

**Rotated MNIST** The 62.000 MNIST images (LeCun et al., 1998) are split into a training, validation and test set of 10.000, 2.000 and 50.000 images respectively, and randomly rotated to orientations between $[0, 2\pi)$. Note that in (Weiler et al., 2018b; Weiler & Cesa, 2019), the rotated MNIST dataset is augmented during training by transforming images with random continuous rotations. We only use train-time augmentation for the state-of-the-art results obtained in Tab. 2, as detailed in App. B.2. All models trained on rotated MNIST, except for the state-of-the-art runs detailed in B.2, are trained for 200 epochs with a batch size of 128 and a learning rate of $1 \cdot 10^{-4}$.

**Scaled MNIST** The 62.000 MNIST images are again split into a training, validation and test set of 10.000, 2.000 and 50.000 images respectively, but now randomly scaled by a factor [0.3, 1] and padded with zeros to retain the original resolution. No data-augmentations of any kind are used in the experiments on scaled MNIST. All models trained on scaled MNIST are trained for 200 epochs with a batch size of 128 and a learning rate of $1 \cdot 10^{-4}$.

**Scaled rotated MNIST** The 62.000 MNIST images are again split into a training, validation and test set of 10.000, 2.000 and 50.000 images respectively, but now randomly scaled by a factor [0.3, 1], padded with zeros to retain the original resolution *and* randomly rotated by orientations between $[0, 2\pi)$. For the experiments on scaled rotated MNIST we do not use data-augmentation of any kind. All models trained on rotated scaled MNIST are trained for 200 epochs with a batch size of 128 and a learning rate of $1 \cdot 10^{-4}$.

**CIFAR10** We evaluate our models on the CIFAR10 dataset, containing 62.000 $32 \times 32$ color images in 10 balanced classes (Krizhevsky et al., 2009). All models trained on CIFAR10 are trained for 200 epochs with a batch size of 128 and a learning rate of $1 \cdot 10^{-4}$.

**CIFAR100** We evaluate our models on the CIFAR100 dataset, containing 62.000 $32 \times 32$ color images in 100 balanced classes (Krizhevsky et al., 2009). All models trained on CIFAR10 are trained for 200 epochs with a batch size of 128 and a learning rate of $1 \cdot 10^{-4}$.

**Galaxy10** This dataset contains 21785 69x69 color images of galaxies divided into 10 unbalanced classes. For Galaxy10, we limit batch size to 32 images, and scale learning rate accordingly to $2.5 \cdot 10^{-5}$, as suggested by Goyal et al. (2017).

**Achieving SOTA on rotated MNIST** To compare performance of our separable regular G-CNNs with previous work, we apply our implementation of $\mathrm{SE}(2)$ and $\mathrm{Sim}(2)$-CNNs to achieve state of the art results on the rotated MNIST dataset, discussed in Sec. 5.

We make some slight adjustment to our architecture and training regime: we reduce the SIREN to a single hidden layer of 64 units. The convolution over $\mathrm{SO}(2)$ is approximated using random sampling, while the convolution over $\mathbb{R}^+$ is approximated using a discretisation with a truncation of the scale group at $s = \sqrt{3}$. We use a batch size of 64, and a learning rate of $1 \cdot 10^{-4}$ and train for 300 epochs. To fairly assess the impact of equivariance, we train a number of models without continuous rotation augmentations.

To compare with previous SOTA, achieved by Weiler & Cesa (2019), we also train models *with* continuous rotation augmentations. Weiler & Cesa (2019) does not specify performance results for models without train-time augmentations, therefore we also compare with the prior SOTA by Weiler et al. (2018b), which trains models both with and without augmentations.

We show results in Tab. 2. With $\mathrm{SE}(2)$-CNNs we achieve best performance with a resolution of 20 elements over $\mathrm{SO}(2)$. For the $\mathrm{Sim}(2)$-CNNs, we achieve best performance with a resolution of 12 elements over $\mathrm{SO}(2)$ and 5 elements over $\mathbb{R}^+$. These results show that without train-time augmentation $\mathrm{Sim}(2)$-CNNs outperform the previous SOTA by Weiler & Cesa (2019), which was trained with continuous rotation augmentations. This improvement further increases when we add train-time augmentation by continuous rotations, to a test error of 0.59%.

**Achieving competitive performance on CIFAR10** To further compare the viability and performance of our separable approach to group convolutions, following Cohen & Welling (2016a) we re-implement the All-CNN-C proposed by (Springenberg et al., 2014), using our separable group convolution layers as drop-in replacement for the regular convolution layers. All experiments described in this section use this exact architecture.

In this experiment we looked to isolate the power of our separable group convolution layers in larger-scale models, which is why we deliberately chose a large but relatively simple architecture. We keep the number of channels constant with the original implementation by (Springenberg et al., 2014). Because of the way our kernels are parameterised, it is hard to make a direct comparison in model performance in terms of representation expressivity, while keeping the number of trainable parameters equal to the original model. Therefore, we train three configurations, with different SIREN hidden sizes.

To obtain a model with approximately the same number of trainable parameters, for the first configuration we limit the hidden size of our SIRENs to 6 to arrive at 1.33m parameters, where the original model has 1.4m. Note that this limitation likely has considerable implications for the expressivity of the sampled group convolution kernels.

To push our separable group convolution layers further, we also train a model configuration with a SIREN hidden size of 5 units, to arrive at a model with 1.14m parameters (19% smaller than the original model by Springenberg et al. (2014)).

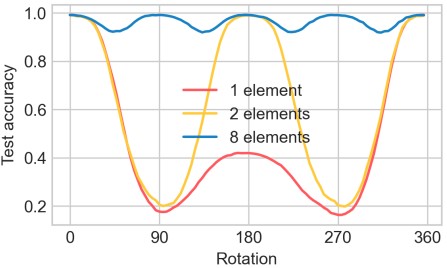 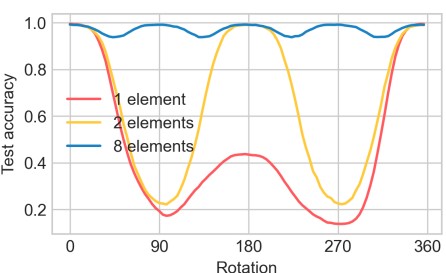

Figure 19: Test error vs rotation angle of MNIST test set, when trained on upright MNIST, for separable SE(2)-CNN.

Figure 20: Test error vs rotation angle of MNIST test set, when trained on upright MNIST, for non-separable SE(2)-CNN.

Lastly, to see how our layers would fare when unimpeded by limitations in SIREN size, we train a configuration with a SIREN hidden size of 16 units. Note that this model contains 3.22m parameters (representing an 128% increase in parameter count to the original model).

All models are trained with a resolution of 8 elements randomly sampled over SO(2) and a discretisation of the scale group of 3 elements truncated at $\sqrt{3}$. We train all of these configurations both with and without data augmentation by padding the original image by at most 6 pixels and randomly cropping to the original resolution, and random horizontal flips. All models are trained for 300 epochs with a learning rate of $1 \cdot 10^{-4}$.

Results are shown in Tab. 3. We can see that for approximately equal parameter counts, our separable group convolution layers equivariant to $\mathrm{Sim}(2)$ outperform the original CNN baseline and the $p4$-G-CNN. Without data augmentation, our model with 6 hidden units is outperformed by the $p4m$-G-CNN which is also equivariant to reflections. With augmentation (containing reflection augmentations) our model outperforms all baseline models we compare to by a significant margin. The smaller configuration with 5 units seems to limit kernel expressivity somewhat, although without data augmentation it still outperforms the translation equivariant original implementation of the All-CNN-C. It seems data augmentation throws the model off in this particular configuration. The larger configuration with 16 hidden units, both with and without data augmentation, significantly outperforms all baselines and other configurations.

## C  ADDITIONAL EXPERIMENTS

### C.1  VALIDATING MODEL INVARIANCE TO TRANSFORMATION GROUPS

Following Weiler et al. (2018b), we empirically assess the level of model-invariance to rotation transformations, by training on upright MNIST, and subsequently assessing performance on a test set of MNIST images that have been transformed by a subgroup element $h \in H$.

Each model is trained on the MNIST training set containing 60,000 images, and tested on 10,000 transformed images. Performance is tested for rotations over a range between $[0, 2\pi)$, in 100 steps. The group convolution is approximated through random sampling. Results are shown in figures 19 and 20 for 1, 2 and 8 group elements for the separable and non-separable variants. We can see that equivariance error is very similar for separable and non-separable variants, and greatly reduces with increased resolution over the group. We do note that for 8 elements, the non-separable group convolution achieves slightly better performance.

### C.2  SUPPORT OF THE SEPARABLE CONVOLUTION KERNELS OVER THE CHANNEL DIMENSIONS

As explained in Sec. 4, we can choose to either define both $k_H$ and $k_{\mathbb{R}^2}$ over the input channel dimension, or incorporate depthwise separability as well; additionally sharing the same spatial kernel $k_{\mathbb{R}^2}$ over the input channels as well as the group input elements. In an additional ablation study we

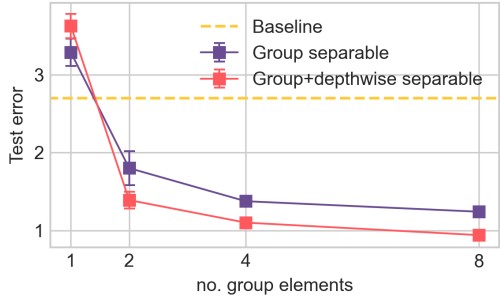 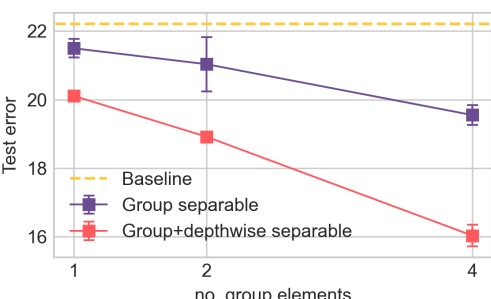

Figure 21: Results for depthwise group separable and group separable $SE(2)-$CNN on rotated MNIST for different resolutions on the group.

Figure 22: Results for depthwise group separable and group separable $\mathbb{R}^2 \rtimes \mathbb{R}^+-$CNN on CIFAR10 for different resolutions on the group.

find that at any resolution above $|H| = 1$, additional depthwise separability outperforms defining both kernels over the input channel dimensions on a fixed parameter budget.

For this experiment on rotated MNIST, we use the same depthwise group separable $SE(2)-$CNN as in Sec. 5, with the architecture described in App. B.1. For the non-depthwise group separable $SE(2)-$CNN, we use a single residual group convolution block of 48 channels instead of two of 32 and 64 channels respectively to arrive at approximately the same number of parameters (815k for the implementation without depthwise separability, 803k for the implementation with depthwise separability). We train the models for 150 epochs, using adam optimisation, with a learning rate of $1 \cdot 10^{-4}$.

We repeat the same experiment for $\mathbb{R}^2 \rtimes \mathbb{R}^+$-CNNs on CIFAR10, for a resolution of 1, 2, and 4 elements. Architectures and training regimes used for the non-depthwise group separable and depthwise group separable experiments are identical to above.

Results, shown in Fig. 21 and Fig. 22 for rotated MNIST and CIFAR10 respectively, highlight that additional depthwise separability outperforms non-depthwise group separable convolutions on a fixed parameter budget, which is why we stick with additional depthwise separation of the spatial kernel, and name this approach the separable group convolution.

### C.3 How does SIREN architecture influence performance?

As explained, we keep the architecture of our SIRENs constant over all experiments. Because we use multiple SIRENs in separable architectures, and we want to isolate the effect of separability of the group convolution operation, we deliberately choose to over-parameterise the SIREN architectures, so as not to advantage the separable implementations by their increased parameter numbers. To this end, we look for the *largest* size of hidden layer that does not negatively impact performance in the non-separable implementation.

We assess the impact of the size of the hidden layers in our SIREN on performance on rotated MNIST for separable and non-separable $SE(2)$-CNN implementations. We sample at a resolution over $SO(2)$ of 8 elements. Interestingly, results in figure 23 show that even with very small SIRENs - a hidden size of 4 or 8 - performance of the non-separable implementations is remarkable. Separable implementations clearly underperform for small SIREN sizes, leading us to hypothesize that the restriction in kernel expressivity due separability paired with under-parameterisation is detrimental to performance in these configurations.

For non-separable implementations, performance barely increases from 16 hidden units on, and starts to decrease after a size of 64 hidden units. For this reason, we choose a size of 64 hidden units in all SIRENs in our experiments.

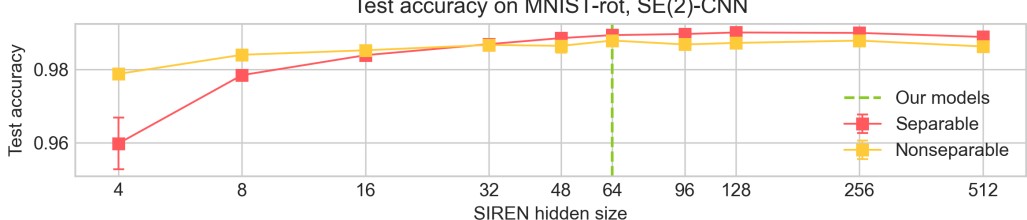

Figure 23: Test accuracy of SE(2)-CNNs on rotated MNIST for different sizes hidden layers in the SIREN parameterisation.

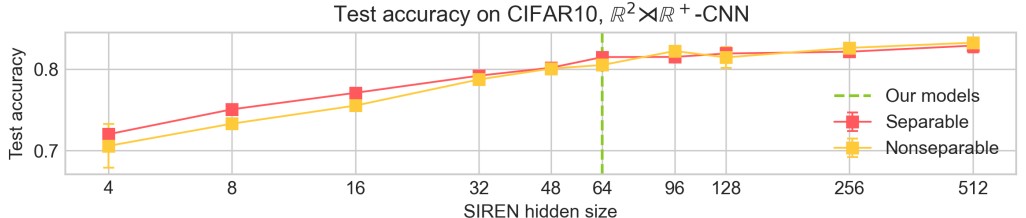

Figure 24: Test accuracy of $\mathbb{R}^2 \rtimes \mathbb{R}^+$-CNNs on CIFAR10 for different sizes hidden layers in the SIREN parameterisation.

We repeat the same experiment for $\mathbb{R}^2 \rtimes \mathbb{R}^+$-CNNs on CIFAR10, and visualise the results in Fig. 24. Here, performance for very small SIREN configurations is less impressive, but we still see a similar saturation around a SIREN size of 64 hidden units.

## C.4 COMPARING ACTIVATION FUNCTIONS FOR CONTINUOUS CONVOLUTION KERNELS

To assess the impact of our use of SIRENs on the performance of our separable G-CNNs, we perform an additional ablation on rotated MNIST in which we compare with other activation functions. We train separable SE(2)-CNNs with a resolution 8 group elements over SO(2), and only vary activation functions in the MLP parameterising the convolution kernels. The convolution on SO(2) is approximated through random sampling.

We look at other activation functions used in parameterising convolution kernels; ReLU (Wu et al., 2019), LeakyReLU, and SiLU (Finzi et al., 2020), see Tab. 5. These results conclusively show the power of sine as activation function, one of the reasons we chose to use SIRENs in our work.

Table 5: Test accuracy (%) of SE(2)-CNNs on rotated MNIST for different activation functions used in parameterisation of the convolution kernel.

| ACTIVATION | SEPARABLE | TEST ACCURACY |
|---|---|---|
| Sine | ✗ | 0.9855 ($\pm$.0012) |
|  | ✓ | **0.9906** ($\pm$.0002) |
| ReLU | ✗ | 0.9721 ($\pm$.0003) |
|  | ✓ | 0.9743 ($\pm$.0034) |
| LeakyReLU | ✗ | 0.9722 ($\pm$.0052) |
|  | ✓ | 0.9788 ($\pm$.0013) |
| Swish | ✗ | 0.9651 ($\pm$.0012) |
|  | ✓ | 0.9595 ($\pm$.0045) |

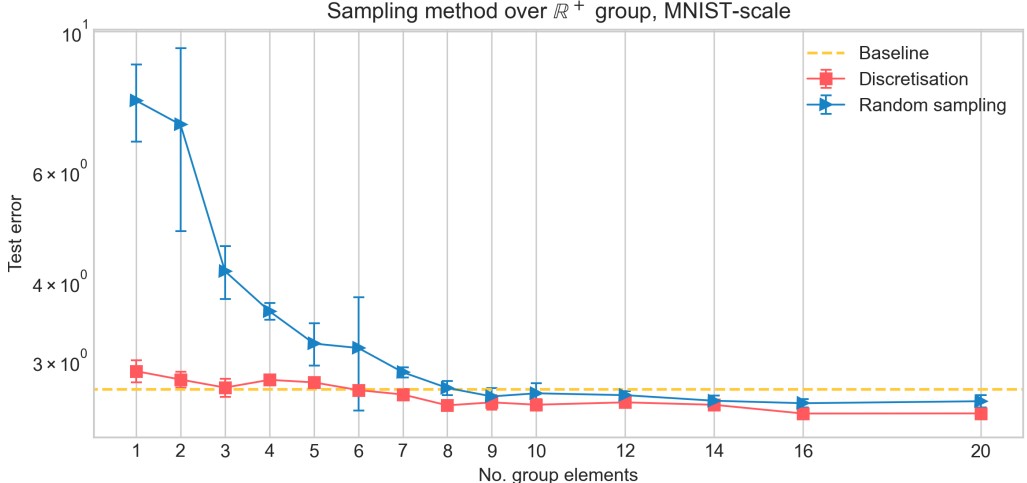

Figure 25: Test error (%) of $\mathbb{R}^2 \rtimes \mathbb{R}^+$-CNNs on scaled MNIST for random sampling versus discretisation of the scale group.

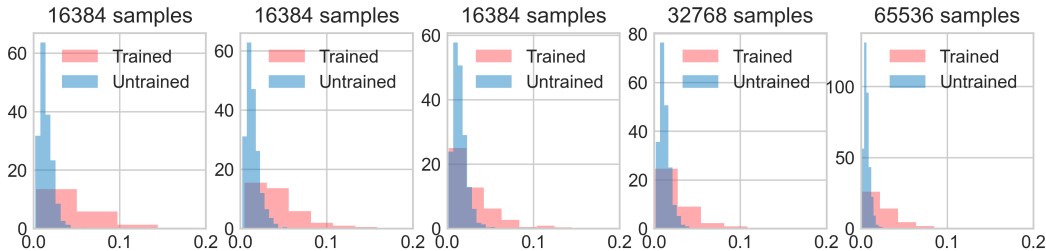

Figure 26: A set of histograms showing variance along the subgroup axis in separable group convolutions. In contrast to Fig. 2, here we show the variability of the convolution kernel along the subgroup axis. This may be seen as an inverse measure of redundancy. Here we can see that contrary to in non-separable group convolution kernels, variability along the group axis increases over the training process, indicating that redundancy decreases. Separable group convolutions thus solve the redundancy issues in group convolution kernels.

## C.5 RANDOM SAMPLING OVER NON-COMPACT GROUPS

As explained in section 4, for non-compact groups we approximate the group convolution through a discretisation of the transformation group of interest. Motivation for this decision is the fact that in non-compact groups, we have to deal with boundary effects of truncating the group, introducing equivariance error into the convolution operation. For dilation, further motivation for approximation through discretisation is the loss of information that occurs in for example natural images, as a result of downscaling a signal on a fixed resolution grid. Random sampling over the dilation group would have as effect that the representation built by a group convolution layer contains different spatial resolutions of information at every sampling step. This results in subsequent layers receiving noisy information at each iteration, which would impede model performance, with training being highly unstable for low resolutions over $\mathbb{R}^+$.

To empirically reinforce this hypothesis, we performed the same experiment with our $\mathbb{R}^2 \rtimes \mathbb{R}^+$-CNN on MNIST-scale described in section 5 with random sampling over the dilation group. Results, shown in figure 25, highlight that indeed, random sampling over the dilation group leads to significantly worse performance. We also noticed great instability during the training process, seen in the variance of the results achieved with random sampling.

### C.6 DO SEPARABLE GROUP CONVOLUTIONS REDUCE REDUNDANCY IN G-CNNS?

As explained, convolving with separable group convolution kernels is analogous to a group convolution with a kernel which shares a reweighting of a single spatial kernel along the group axis. This approach was motivated by redundancy observations in regular group convolution kernels along the group axis, seen in Fig. 2.

To assess whether the separable group convolution solves these redundancy issues in G-CNNs, we may want to perform a similar test of the variability of our introduced separable implementation along the group axis. Note that, if we were to reconstruct the full group convolution kernel $k$ from $k_H$ and $k_{\mathbb{R}}^2$ and apply the same PCA variance test as used to obtain Fig. 2, we would find that each group convolution could be fully explained by the first principal component (corresponding to the reshared spatial kernel). Instead, we measure the variability of our separable kernels by assessing the variance of $k_H$, which lies along this axis. Results are shown in figure 26 for the separable SE(2)-CNN trained on Galaxy10 dataset with a resolution of 8 elements on SO(2). Here, we observe the variance increasing in all layers over the training process. This indicates that indeed, separable group convolutions reduce parameter redundancy in G-CNNs.

