# OpenReview forum: "Exploiting Redundancy: Separable Group Convolutional Networks on Lie Groups"
_ICLR.cc/2022/Conference — ICLR 2022 Submitted_

### Official Review · Reviewer_4A1c · 2021-10-29

**Correctness:** 4
**Technical Novelty And Significance:** 2
**Empirical Novelty And Significance:** 2
**Recommendation:** 5
**Confidence:** 4

**Main Review:**

*Strengths*

S1: The paper is well-written, the method introduced makes sense and seems to work well in practice.

*Weaknesses*

W1: The submission has somewhat limited novelty. It can be seen as an application of depthwise convolutions to group-CNNs. Depthwise convolutions are quite popular since at least 2017 (Xception, Mobilenets); and the (non-separable) group-CNN utilized seems the same as Finzi et al [1], with sinusoidal activations. More importantly, the idea of separating the group-convolution along the group dimension has also appeared in Lengyel and van Gemert [2], although for a different flavor of G-CNN.

W2: I see several issues with the experimental section. The most problematic is that all comparisons are against the paper's own baselines. While this makes for fair comparisons in the sense that the data pipeline, architectures and training schedule are all the same (although I have reservations about the number of parameters, see W3), I'm not sure if the conclusions hold in general. For example Weiler and Cesa [3] seems to show significantly better performance than the proposed method on rotated MNIST and CIFAR10/100; can the proposed method improve those results? If not, what would be the applications where the method is useful? Finzi et al [1] also show favorable results on a molecular property prediction, how does the submission fare in that task? When comparing with Finzi et al, I believe there should also be an ablation to disentangle the effects of the separable convolutions and the use of SIREN instead of a regular MLP.

W3: Please report the number of parameters for each model trained. If I understand correctly, the number of channels per layer is kept constant, so the non-separable models have several times more parameters than the separable, is that correct? In that case could the lower performance on non-separable be explained by overfitting or slower convergence? I think both constant number of channels per layer or constant number of parameters are informative and should be reported (Weiler and Cesa [3] do something similar for CIFAR).

W4: In Table 5, I believe a few experiments are missing and would be needed to disentangle the effects of the separation along the group dimension and the channel dimensions. I suggest to show, for each group, the performance when the convolutions are separable over the channel (depthwise) but not the group dimension. And for the baseline, it would be interesting to also see the performance for the separable depthwise version.

*Questions*

Q1: For the rotated MNIST, it is shown that approximating the convolution with random samples is superior, however the other MNIST variations seem to fall back to the discretization. Why is this the case? Do the random sampling approximation perform worse on the higher dimensional groups?

Q2: The SIM(2) MNIST experiment is described as limited to 2, 4, 6, or 8 elements for each subgroup. Does this also refer to the way the dataset is constructed, or is it created with random rotations and scaling sampled from the continuous internal?

*References*

[1] Finzi et al, "Generalizing convolutional neural networks for equivariance to lie groups on arbitrary continuous data", ICML'20.

[2] Lengyel and van Gemert, "Exploiting Learned Symmetries in Group Equivariant Convolutions", ICIP'21.

[3] Weiler and Cesa, "General E(2) - Equivariant Steerable CNNs", NeurIPS'19.

**Summary Of The Paper:**

The paper proposes to use separable convolutions along the group dimension in the type of group CNNs proposed by Finzi et al [1]. The motivation is the same as the popular depthwise separable convolutions for conventional CNNs: reducing parameter redundancy to increase efficiency and accuracy. Experiments on rotated MNIST, CIFAR10/100, and Galaxy10 show that the method outperforms the non-separable versions both in accuracy and speed.


**Summary Of The Review:**

I believe the paper is not there yet, mainly because of limited novelty when compared to [1, 2] (W1), lack of comparison against the state-of-the-art of the tasks proposed (W2), and not very convincing experimental results due to lack of details (W3) and ablations (W4).

---

> ### Author Response · Authors · 2021-11-18
> **Response to reviewer 4**
>
> First off, we would like to thank you for your thorough assessment of our work. We feel that your many insightful comments and suggestions helped us make our paper a stronger contribution to the field of equivariant neural networks. We address each of your concerns below:
>
> > 1. The submission has somewhat limited novelty. It can be seen as an application of depthwise convolutions to group-CNNs. Depthwise convolutions are quite popular since at least 2017 (Xception, Mobilenets); and the (non-separable) group-CNN utilized seems the same as Finzi et al [1], with sinusoidal activations. More importantly, the idea of separating the group-convolution along the group dimension has also appeared in Lengyel and van Gemert [2], although for a different flavor of G-CNN.
>
> We understand the reviewers' concerns. Indeed, the current approach builds upon works by Cohen & Welling (2016), Bekkers, (2020) and Finzi (2020). The reason we feel this work is relevant is that, to the best of our knowledge, all existing approaches to regular group convolutions suffer from important computational restrictions when handling large groups. In particular, the number of samples required in order to execute group CNNs with regular representations on large groups grows exponentially with the independent subgroups in the group. For instance, if we consider 4 elements per dimension on the Sim(2) group (composed of rotations & scaling), the number of elements required for the approximation becomes 16. (4 rotations for each of the 4 scales considered in the approximation.  This major drawback is what keeps group CNNs from being used in very large groups.
>
> Our separable G-CNNs address precisely this issue. As shown in our work, the separable group convolution allows for the creation of G-CNNs equivariant to very large groups, which grow linearly in computational complexity in the number of dimensions of the group (as opposed to exponentially). This allows us to create G-CNNs equivariant to the Sim(2) group. To the best of our knowledge, Separable G-CNNs are the first work in providing equivariance to this large group.
>
> To the best of our knowledge, Lengyel & van Gemert (2021) is the only existing work, which tackles the problem of separability in the group equivariant setting. However, they only consider the p4m and p4 groups considered in  Cohen & Welling (2016), and subsequently their approach only works for discrete groups where the group action on the convolution kernel can be expressed by a permutation of the kernel grid (a highly restrictive condition). Our work presents a flexible framework for separable group equivariance that generalises to arbitrary compact and non-compact affine Lie groups.
>
> Moreover, we find some considerable differences between depthwise separable convolutions (Haase & Amthor, 2020; Chollet, 2016) and our work. Depthwise separable convolutions essentially separate (a) information mappings in the domain of the data (combining information on spatial configurations of features) from (b) information mappings in the signal itself (combining different features at a single spatial location). separable group convolution kernels, on the other hand, separate information mappings in the domain of the data by (1) first combining information on features under different poses (orientations in case of SE(2)) at a given spatial location, and (2) combining observed features at different spatial locations next. Contrary to Haase & Amthor (2020), Chollet (2016), these convolution operations perform information mappings that split the domain of the data, i.e., the group, to improve computational performance. In addition, we would like to emphasize that ensuring group equivariance while separating its domain is a non-trivial problem, and our method provides a general framework for this purpose..

---

> > ### Author Response · Authors · 2021-11-18
> > **Response to reviewer 4 (part 4)**
> >
> >
> > > Q1. For the rotated MNIST, it is shown that approximating the convolution with random samples is superior, however the other MNIST variations seem to fall back to the discretization. Why is this the case? Do the random sampling approximation perform worse on the higher dimensional groups?
> >
> > For any experiments with the compact (sub-)group $SO(2)$, we approximate $SO(2)$ using random sampling. For $\mathbb{R}^2$ we use a discretisation, as our input images are defined on a discretisation of $\mathbb{R}^2$ (and since, importantly, this allows us to use pytorch convolution implementations). Lastly, for $\mathbb{R}^+$, we truncate the group, and discretise it. For equivariance to $\mathrm{Sim(2)}$, this means that the group convolution is approximated with randomly sampled rotation elements, and discretized translation and dilation elements. These details can also be found in section 4 (Approximating equivariance for compact continuous and non-compact continuous groups).
> >
> > Although we experimented with random sampling over the dilation group, we found this to be highly unstable (we added the results for random sampling of the scale group to Appx C.5). Intuitively, the dilation group is unlike the rotation group in two aspects; it is non-compact, meaning we have to account for boundary effects that occur, and its action on natural images with a fixed spatial resolution causes information loss when traversing the dilation group from larger to smaller scale values. Essentially, the decrease in spatial resolution for smaller scales equates to a smoother function at the original resolution. In this setting, we thus conjecture that due to the changing spatial resolution when traversing the group, random sampling may have too strong of a regularising effect on the learned kernels, to the point that the network is unable to build sufficiently expressive representations.
> >
> > > Q2. The SIM(2) MNIST experiment is described as limited to 2, 4, 6, or 8 elements for each subgroup. Does this also refer to the way the dataset is constructed, or is it created with random rotations and scaling sampled from the continuous internal?
> >
> > The rotated scaled mnist dataset is created from continuous random rotations between 0 and 2pi and continuous random dilations between 0.3 and 1, we expanded on the experimental section to make this clearer.
> >
> > _Weiler, M., & Cesa, G. (2019). General $ E (2) $-Equivariant Steerable CNNs. arXiv preprint arXiv:1911.08251_
> >
> > _(Cohen, T., & Welling, M., 2016)  “Group equivariant convolutional networks” ICML’16_
> >
> > _(Finzi, M., Stanton, S., Izmailov, P., & Wilson, A. G. 2020). “Generalizing convolutional neural networks for equivariance to lie groups on arbitrary continuous data”. ICML’20._
> >
> > _(Bekkers, E. J. 2019). “B-spline cnns on lie groups”. ICLR’20_
> >
> > _(Lengyel, A., & van Gemert, J. C. 2021). Exploiting Learned Symmetries in Group Equivariant Convolutions. ICIP’21._
> >
> > _(Haase, D., & Amthor, M. 2020). “Rethinking depthwise separable convolutions: How intra-kernel correlations lead to improved MobileNets”. CVPR’20_
> >
> > _(Chollet, F. 2017). “Xception: Deep learning with depthwise separable convolutions.” CVPR’17_

---

> > ### Author Response · Authors · 2021-11-18
> > **Response to reviewer 4 (part 3)**
> >
> >
> > > 3. Please report the number of parameters for each model trained. If I understand correctly, the number of channels per layer is kept constant, so the non-separable models have several times more parameters than the separable, is that correct?
> >
> > Indeed, we keep the number of channels in our original experiments constant, again, in order to isolate the impact of separating the group convolution kernel. However, note that our convolution kernels are parameterised by SIRENs, decoupling the number of trainable parameters from the sampled resolution over G. in the non-separable case we have a single SIREN $\frak{g}$ $\rightarrow \mathbb{R}^{C_{in} \times C_{out}}$, whereas in the separable case, as we are now parameterising two distinct convolution kernels, we have two distinct SIRENs. Assuming only $k_H$ extends over the input channels as we do in our experiments, the first SIREN $\frak{h}$ $\rightarrow  \mathbb{R}^{C_{in} \times C_{out}}$ and the second $\mathbb{R}^2 \rightarrow  \mathbb{R}^{C_{out}}$ (these details can also be found in Appx. B.1 (architecture details)). This means the separable implementation actually contains more trainable parameters, (approx 7.5% more parameters, since the second SIREN is relatively small compared to the first). We do feel we need to point out that the number of trainable parameters in our implementation is not a good indicator of the expressivity of the learned representations throughout the network. As explained, separable group convolution kernels are strictly less expressive than non-separable group convolutions. As such, we chose to keep the number of channels constant throughout the network, as we feel this number bears more relevance to the expressivity of the representations formed by our networks, and over-parameterise our SIRENs, as shown in App C.3.
> >
> > > In that case could the lower performance on non-separable be explained by overfitting or slower convergence?
> >
> > Interesting question., We conjecture that separating the group convolution operation has regularising effects on the learned feature representations throughout the network. For example, in the rotated MNIST experiments we notice that, for the non-separable configurations, as we keep increasing the sampling resolution over the group, the test set performance actually decreases (we did not notice significant differences in convergence for different sampling resolutions). As we discussed in section 5 (Rotated MNIST) this is in line with findings by Bekkers (2019), who note that as the group resolution increases, so does the possibility for overfitting on specific spatial configurations. However, we can see that no such drop in performance happens as we increase the resolution in the separable case, indicating that indeed, the reduced kernel expressivity in separable group convolutions may prevent overfitting on such spatial configurations of features.
> >
> > > I think both constant number of channels per layer or constant number of parameters are informative and should be reported (Weiler and Cesa [3] do something similar for CIFAR).
> >
> > We added details on the number of trainable parameters in our setups to Appx. B.1. and rewrote the introduction to our experiments section to more clearly highlight our choice of keeping the number of channels constant throughout our experiments.
> >
> > > 4. In Table 5, I believe a few experiments are missing and would be needed to disentangle the effects of the separation along the group dimension and the channel dimensions. I suggest to show, for each group, the performance when the convolutions are separable over the channel (depthwise) but not the group dimension. And for the baseline, it would be interesting to also see the performance for the separable depthwise version.
> >
> >
> > We understand and share the reviewers’ concerns about the effects of additional depthwise separation. To this end, our original submission contained an additional ablation study in App. C.2, showing how, on a fixed parameter budget, additional depthwise separation of the separable group convolution consistently outperforms models without additional depthwise separation on rotated MNIST. To reinforce these results, we performed the same experiment without depthwise separation on CIFAR10, and added these to App. C.2 as well, showing similar results.
> >
> > Furthermore, for the experiments on scaled MNIST, we also experimented with separable implementations with a single group element. As this experiment uses a discretisation of the scale group, meaning the same scale element is sampled at every layer, this configuration equates to a regular CNN with depthwise separation. The non-separable implementation using a discretisation of 1 group element equates to a regular CNN without depthwise separation. We see that in this setting, depthwise separation actually considerably impedes model performance. We feel depthwise separation itself has been investigated thoroughly in works like Chollet (2016) Haase & Amthor (2020).

---

> > ### Author Response · Authors · 2021-11-18
> > **Response to reviewer 4 (part 2)**
> >
> >
> > > 2. I see several issues with the experimental section. The most problematic is that all comparisons are against the paper's own baselines. While this makes for fair comparisons in the sense that the data pipeline, architectures and training schedule are all the same (although I have reservations about the number of parameters, see W3), I'm not sure if the conclusions hold in general. For example Weiler and Cesa [3] seems to show significantly better performance than the proposed method on rotated MNIST and CIFAR10/100; can the proposed method improve those results? If not, what would be the applications where the method is useful? Finzi et al [1] also show favorable results on a molecular property prediction, how does the submission fare in that task?
> >
> > As the reviewer points out, we intentionally selected a simple architecture for our experiments to isolate the effect of separating the group convolution kernel, and fairly assess the impact of sampling resolution over subgroup $H$. Indeed, better architectures for the CIFAR10 task exist, but contain more components (e.g. augmentation) that could potentially add noise to our assessments about separating the group convolution operation. Still, we appreciate the need for comparison with other approaches, as this gives a more absolute representation of the capabilities of our models. To this end, we added a section showing how separable group convolutions allow our $\mathrm{Sim(2)}$-CNN to achieve SOTA on rotated MNIST, considerably improving upon previous results by Weiler et al. (2019), with minimal changes to our experimental setup.
> >
> > *edited*: Furthermore, we reimplemented the All-CNN-C architecture used in Cohen & Welling (2016) for CIFAR10, using our separable group convolutions as drop-in replacement for the regular convolutional layers. We show that our $\mathrm{Sim(2)}$ convolutional layers improve performance of the original All-CNN-C, and all versions of the G-CNN proposed by Cohen & Welling (2016), on a similar parameter budget.
> >
> > Also, we are currently working on implementing separable SO(3)-CNNs and applying them to molecular data, e.g., the QM9 dataset, to compare with Finzi (2020). **edit: see follow-up comment for details!**
> >
> > > When comparing with Finzi et al, I believe there should also be an ablation to disentangle the effects of the separable convolutions and the use of SIREN instead of a regular MLP.
> >
> > We understand the concerns about the impact of the sine activation function on model performance, and add an additional ablation study comparing activation functions for separable and non-separable architectures on rotated MNIST in App. C.4. This experiment shows conclusively that kernels parameterised with SIRENs outperform ReLU, LeakyReLU and SiLU MLPs. However, we would like to point out that our choice of activation function has no impact on the validity of our conclusions on separation of the group convolution; all our implementations, separable and non-separable, are implemented with sine activations.

---

> > > ### Author Response · Authors · 2021-11-22
> > > **Follow-up: applying separable G-CNNs to higher-dimensional domains**
> > >
> > > We have implemented a basic ResNet type point convolution architecture for molecular property prediction on the QM9 dataset and ran 3 versions of it: (i) a standard R3 network (Conv3D), (ii) an R3 network with isotropic kernels such that rotation equivariance is guaranteed, (iii) a fully SE(3) equivariant network via separable group convolutions. The net has a point-wise embedding layer, 3 resnet blocks ([conv + batch-norm + act_fn]x2 + skip), and a pooling with prediction layer. It thus 6 convolution layers in total.
> > >
> > > Considering the time constraints, we have only managed so far to run the experiments for predicting the alpha value, and only trained for 250 epochs. The results confirm what we expect, namely that equivariance constraints are important and are easily implemented with our method. As is standard, we evaluate using the mean absolute error MAE on the predicted property:
> > > - R3 Convs: MAE = 0.310
> > > - R3 Convs (isotropic kernels): MAE = 0.215
> > > - G-Convs (SE(3) equivariant): MAE = 0.142
> > >
> > > Since these results are quite preliminary and could benefit from more parameter tuning and architecture searches, as well as a full evaluation on all properties, we would like to include these results and accompanying code as a use case in our GitHub repository, rather than including it in the paper as formal results. If the reviewers deem appropriate, however, we would still be happy to include in the paper as preliminary results in an appendix, referring to it in the main body as extra empirical evidence in favor of our work.
> > >
> > > It was precisely the separability and the flexibility of working with continuous kernel parametrization that allowed us to easily extend the work to 3D translations + rotations. The separability was particularly important in the QM9 point cloud type dataset, as there graph NN had to be used to handle the varying neighborhoods. We could not fit a naïve full group conv implementation in memory (the neighborhoods get too large in the non-sep case), whereas this was no problem for the separable implementation. We worked with random, pseudo-uniform grids of 12 rotations in SO(3), obtained by maximizing the geodesic distance between rotation matrices in the grid.
> > >
> > > Finally, regarding the significance of these numbers. Our preliminary results are in the ball-park of methods such as tensor field networks (MAE=0.223, Thomas et al.) and SE(3)-transformers (MAE=0.142, Fuchs et al.). The state-of-the-art has predictions in the range MAE 0.04-0.07. We expect that with further tuning we could push our results towards SOTA, but given the timeline we were very happy to see that with minimal effort we could already significantly improve on classic point conv approaches (note early works such as Schnet (Kristof et al.) also use isotropic 3D convolutions like our case (ii) with which they got to an MAE=0.235).
> > >
> > > _references_
> > >
> > > _Thomas, N., Smidt, T., Kearnes, S., Yang, L., Li, L., Kohlhoff, K., & Riley, P. (2018). Tensor field networks: Rotation-and translation-equivariant neural networks for 3d point clouds. arXiv preprint arXiv:1802.08219._
> > >
> > > _Fuchs, F. B., Worrall, D. E., Fischer, V., & Welling, M. (2020). Se (3)-transformers: 3d roto-translation equivariant attention networks. arXiv preprint arXiv:2006.10503._
> > >
> > > _Schütt, KT, Sauceda, HE, Kindermans, PJ, Tkatchenko, A., & Müller, KR (2018). Schnet - a deep learning architecture for molecules and materials. The Journal of Chemical Physics , 148 (24), 241722._

---

> ### Comment · Reviewer_4A1c · 2021-11-27
> **Response to rebuttal**
>
> Thanks for the extra material included in the paper, I do think the submission is stronger now. However, many of my concerns remain. My main issues are with respect to i) novelty and ii) how much the novel contributions are contributing to the improved performance.
>
> i) I understand that novelty might be hard to assess, but in my opinion the submission falls short on this regard for the reasons mentioned before. It might still be worth publishing since the findings are potentially useful for the community.
>
> ii) If I understand correctly, the differences between the submission and Finzi et al (2020) are 1) use of separable group convolution, 2) use of depthwise separable convolution, 3) use of SIREN to parametrize the filters, 4) architectures/training schedule. The novel contribution is 1) so I believe the experiments should be designed to demonstrate how much improvement does that bring. This is why I suggested:
>
> a) running experiments with only depthwise separable convolutions (this could have been shown in Figs 21, 22).
>
> b) running experiments without SIREN -- this has been shown in Table 5 but I'm puzzled because the model with Swish, which should be close to Finzi et al (2020), does significantly worse than their published results (probably due to differences in architectures/training schedule). This is a manifestation of the problems of comparing against the paper's own baselines, which reduces my confidence in the results. I think that the group-separable convolutions may cause a drop in accuracy (while improving efficiency) but the use of SIREN compensates for that. It is hard to disentangle the effects with the current experiments.
>
> Anyway, I think this is a borderline paper. I wouldn't oppose neither acceptance or rejection.

---

### Official Review · Reviewer_xJAv · 2021-11-02

**Correctness:** 4
**Technical Novelty And Significance:** 4
**Empirical Novelty And Significance:** 1
**Recommendation:** 6
**Confidence:** 3

**Main Review:**

This is a solid paper to explore the combination of group convolutions and techniques employed in popular CNN architectures. The ideas of how we can achieve $Sim(2)$-equivariant on separable convolution operations are clearly explained, they are helpful in designing kernels with better performance. Besides, the paper itself is educational to readers not familiar with this field, I recommend accepting this paper to let more researchers benefit from the ideas behind G-CNNs.

I have a few questions to ask:

1. The motivation is explained as the group convolutions learn redundancies. Is this only found in G-CNNs or all types of CNNs? Do separable convolution operations address this problem finally? Can we get a figure similar to Figure 2?

2. Although the utilization of SIREN is the highlight of the paper, the motivation behind is unexplained and unclear. Why the group convolution kernels can be represented implicitly? Is SIREN the only choice?  Do larger SIREN networks provide better parameterization?

3. Why does random sampling over subgroups provide better performance? I am baffled by these results, if we can design a discretization sampling method that has similar anti-aliasing effects?

**Summary Of The Paper:**

The paper builds group convolutional neural networks based on the depth-wise separable convolution operations, which are commonly seen in modern CNNs. The authors demonstrate that the $Sim(2)$-equivariant can be achieved in such separable convolution operations. In the implementation, the authors borrow the SIREN approach for parameterizing the proposed group convolutions. Finally, the experimental analysis shows good improvement over other types of group convolutions.

**Summary Of The Review:**

The bad thing of such mathematical ideas inspired paper is that the experimental performance are always far from sota. But I think the performance is not the only thing, it is not bad to accept this paper for me.

---

> ### Author Response · Authors · 2021-11-18
> **Response to reviewer 3**
>
> Dear reviewer 3, thank you for your thorough review and your insightful and interesting comments and questions. We are glad you enjoyed the read, and share your hope that this paper and project may serve as an introduction into the field of equivariant neural networks for many. To further strengthen the readability and understandability of our paper, we added some visualisations of relevant operations to the appendix..
>
> We will now address your comments:
>
> > 1. The motivation is explained as the group convolutions learn redundancies. Is this only found in G-CNNs or all types of CNNs?
>
> Interesting question! Similar types of redundancy indeed occur in conventional CNNs; Sifre (2014), Chollet (2017), Haase & Amthor (2020) experimentally show redundancy in learned convolution kernels along the channel dimension. However, group convolution kernels are different in that their domain consists of multiple subgroups, for example in the case of SE(2) two spatial dimensions and one rotation dimension (whereas the kernels of regular 2D CNNs generally only extend over the spatial domain). The observed redundancy in group convolution kernels occurs along the non-spatial subgroup axis (the rotation axis in the case of SE(2)). Depthwise separable convolutions essentially separate (a) information mappings in the domain of the convolution kernel (combining information on spatial configurations of features) from (b) information mappings in the co-domain of the kernel (combining different features at a single spatial location). On the other hand, separable group convolution kernels (also) separate information mappings in the domain of the kernel, (1) first combining information on features under different poses (orientations in case of SE(2)) at a given spatial location, then (2) combining observed features at different spatial locations. Both these convolution operations thus perform information mappings in the co-domain of the kernel.
>
> > Do separable convolution operations address this problem finally? Can we get a figure similar to Figure 2?
>
> Separable group convolutions address the observed redundancy by explicitly sharing (a reweighting of) the same spatial kernel along the group axis to create a group convolution kernel (we added a figure in the appendix explaining this intuition). Since the kernel along the group axis is now fully explained by a single spatial kernel, if we apply the same experiment as for Fig. 2, we find that the first principal component fully explains the variance along the group axis (the first principal component now corresponds exactly to the spatial kernel, and the values of $k_H$ correspond to the coordinates in this new basis). However, by inspecting the variance of sampled convolution kernels on the subgroup $H$, we may still draw some conclusions on the degree of redundancy in our proposed separable group convolution parameterisation. We added an additional figure to the appendix (Fig. 15) similar to Fig. 2, showing the variance along the subgroup axis in the separable kernel increases during the training process, indicating that indeed, separable group convolutions reduce redundancy in G-CNNs.

---

> > ### Author Response · Authors · 2021-11-18
> > **Response to reviewer 3 (part 2)**
> >
> > > 2. Although the utilization of SIREN is the highlight of the paper, the motivation behind is unexplained and unclear. Why the group convolution kernels can be represented implicitly? Is SIREN the only choice?
> >
> > We understand the reviewers’ concerns about the motivation behind our choice of parameterisation. Nevertheless, we would like to emphasize that the goal of our paper, and thus its highlight is the formulation of separability in the group equivariant setting.
> >
> > However, the parameterization is an important point of our work, which we failed to address correctly in our work. We failed to discuss the increased expressivity that sine activations bring. Sine activations or similar high-frequency inducing functions applied to the data allow MLPs to learn high-frequency functions on low-dimensional domains, where ReLU’s have been shown to underperform (Sitzmann 2020), (Tancik 2020). We expanded on section 4 (Defining convolution kernels on Lie algebras), to include these details. Furthermore, we added an experiment to the appendix comparing performance using SIRENs as kernel parameterisation versus using MLPs with ReLU, LeakyReLU or SiLU, which reinforces findings by Romero (2020) with regards to kernel expressivity.
> > Lastly, our original submission only briefly mentioned another property of SIRENs that makes them a desirable kernel parameterisation for these group convolution kernels; their explicit control over the smoothness of the learned function through $\omega_0$. We added a section to the appendix visually explaining the importance of kernel smoothness in G-CNNs for continuous groups.
> >
> > > Do larger SIREN networks provide better parameterization?
> >
> > We have now added an ablation to the appendix, showing the impact of the size of the SIREN architecture on the performance of the resulting G-CNN. This experiment shows that (1) even very small SIREN architectures lead to remarkable results and (2) beyond a certain size of SIREN, performance starts to degrade. Note that for all our experiments, we picked the largest SIREN that did not negatively impact performance on the non-separable implementations, to fairly compare the performance difference between separable and non-separable G-CNNs purely on separation of the group convolution.
> >
> > > 3. Why does random sampling over subgroups provide better performance? I am baffled by these results, if we can design a discretization sampling method that has similar anti-aliasing effects?
> >
> > We think the explanation for the improvement in performance for random sampling over compact groups is twofold. First, through random sampling we obtain an unbiased estimation of the group convolution operation, whereas a discretisation biases the network to be equivariant only to a subset of the continuous group (see Wu et al, (2019) for details in this regard).
> >
> > Secondly, we think that random sampling over the group has a regularising effect on frequencies contained in the learned kernels. As the full group is sampled, the analytical kernel function is learned over this entire group, meaning it learns to transition smoothly between neighbouring group elements, possibly reducing aliasing effects.
> >
> > In the case of (coarse) static discretization, aliasing is not a problem. This is because the MLP will be always looking at the same points, and thus, (although the network could be intrinsically aliased), the convolution would not see its effect.
> >
> > However, If we would go from a small number of group elements (say 4) to more elements (say 16) with the same network, then aliasing might appear and might be detrimental for performance. By always sampling different points of the group, the network learns to be smooth at all positions and thus, such an approach could be applicable (for instance to fasten up training).
> >
> > _References_
> >
> > _Wu, W., Qi, Z., & Fuxin, L. (2019). Pointconv: Deep convolutional networks on 3d point clouds. In Proceedings of the IEEE/CVF Conference on Computer Vision and Pattern Recognition (pp. 9621-9630)._
> >
> > _(Sitzmann, V., Martel, J., Bergman, A., Lindell, D., & Wetzstein, G. 2020). “Implicit neural representations with periodic activation functions”. NIPS’20_
> >
> > _(Tancik, M., Srinivasan, P. P., Mildenhall, B., Fridovich-Keil, S., Raghavan, N., Singhal, U., ... & Ng, R. 2020). Fourier features let networks learn high frequency functions in low dimensional domains._
> >
> > _(Romero, D. W., Kuzina, A., Bekkers, E. J., Tomczak, J. M., & Hoogendoorn, M. 2021). CKConv: Continuous Kernel Convolution For Sequential Data._
> >
> > _(Haase, D., & Amthor, M. 2020). “Rethinking depthwise separable convolutions: How intra-kernel correlations lead to improved MobileNets”. CVPR’20_
> >
> > _(Chollet, F. 2017). “Xception: Deep learning with depthwise separable convolutions.” CVPR’17_
> >
> > _(Sifre, L., & Mallat, S. 2014). Rigid-motion scattering for texture classification, PhD Thesis_

---

### Official Review · Reviewer_m8cZ · 2021-11-03

**Correctness:** 2
**Technical Novelty And Significance:** 2
**Empirical Novelty And Significance:** 1
**Recommendation:** 5
**Confidence:** 5

**Main Review:**

The reviewer is very interested in this research work. This paper explores the shortcomings of existing work in depth, discusses separable convolution kernels, and does the parameterization of different groups more comprehensively, which  is a good research guide. In addition, the authors also conducted experiments on many vision datasets to illustrate the superiority of the proposed method and achieved performance improvements.

However, there exist some confusing parts or weaknesses to be further strengthened.
1. The biggest shortcoming of this paper is its limited innovation. As stated in this paper, the proposed method is more like a promotion of Cohen2016, by using a strategy similar to ‘depthwise separable convolution’ in Xception. It seems that this work depends on the existing method heavily and its original parts bring weak contributions.

2. The theoretical foundation of this paper is not solid. It is worth mentioning that the authors explored arbitrary affine Lie groups (three different groups: SE2, dilation +translation, and dilation +Sim(2)). But as far as the reviewer knows, this is not entirely applicable. For example, correlation is an equivariant map for the translation but NOT for the rotation group. Why can the authors directly consider SE(2) and Sim (2) in convolution (both containing Rotation)? In addition, when proving translation equivariant, it is necessary to extend f and k to group G, otherwise g^(-1)x does not belong to group G and cannot go on. The reviewer has not seen relevant explanations. Regarding Sim(2), it has a totally different Lie algebra expression from SE(2)’s due to the scaling, and there has not existed a discussion on specific forms.

3. The motivation of the proposed method comes from parameter redundancy, but from Figure 1, the reviewer found that the metric is F-test (variance ratio), which is usually used to analyze statistical models that use more than one parameter to determine whether all or part of the parameters are suitable for estimating the population. And the robustness of F-test is extremely dependent on the samples with a normal distribution. In the case of limited samples, the distribution is often biased, which tends to reduce the power of such a test, leading to incorrect statistical inference.

4. The authors used SIREN as kernel parameterization in the Lie algebra. But as far as the reviewer knows, this work proves the superiority of using periodic activation functions (such as sine), which is, however, irrelevant to this paper from the perspective of the reviewer. Much more details should be provided here.

5. Another concern of the reviewer is that evaluation results of this paper are not convincing to the reviewer. Traditional CNN-based methods have achieved 95+% accuracy on CIFAR10, and regular GNNs (p4m & Z2) can also reach 90+%. The accuracy results of this paper are still inferior to 90% with a 77% baseline. It violates the claim that "G-CNNs usually improve upon regular CNNs" stated in this paper.

6. Some parts are not novel at all, such as the section “Depthwise separability”. The reviewer suggests the authors to include more relevant references (Xception or more).


**Summary Of The Paper:**

This paper discusses considerable parameter redundancy in regular group convolution networks and then proposes separable convolution kernels to share the weights over the subgroups. Besides, the authors explored the equivariance of three different groups and presented a continuous parameterization scheme. Evaluations on several datasets show that their proposed method gains improved performance and computational efficiency.

**Summary Of The Review:**

The research field of this paper is very interesting, but unfortunately there remain many shortcomings in theoretical and experimental results, which need to be forcefully addressed. The reviewer therefore thinks that the paper is not good enough to be published on ICLR.

After reading the authors' rebuttal, I increase my rating a bit. But I am still not contented with the stuff in this submission, which should undergo a thorough revision to convince and be easily understood by the readers.

---

> ### Author Response · Authors · 2021-11-18
> **Response to reviewer 2**
>
> Dear reviewer 2, we thank you for your thorough review of our work, and for the many valuable suggestions and criticisms given. We will address each of them individually below.
>
> > 1. The biggest shortcoming of this paper is its limited innovation. As stated in this paper, the proposed method is more like a promotion of Cohen2016, by using a strategy similar to ‘depthwise separable convolution’ in Xception. It seems that this work depends on the existing method heavily and its original parts bring weak contributions.
>
> We appreciate the reviewers' concerns. Indeed, the current approach builds upon works by Cohen & Welling (2016), Bekkers, (2020) and Finzi (2020). The reason we feel this work is relevant is that, to the best of our knowledge, all existing approaches to regular group convolutions suffer from important computational restrictions when handling large groups. In particular, the number of samples required in order to execute group CNNs with regular representations on large groups grows exponentially with the independent subgroups in the group. For instance, if we consider 4 elements per dimension on the Sim(2) group (composed of rotations & scaling), the number of elements required for the approximation becomes 16. (4 rotations for each of the 4 scales considered in the approximation).  This major drawback is what keeps group CNNs from being used in very large groups.
>
> Our separable G-CNNs address precisely this issue. As shown in our work, the separable group convolution allows for the creation of G-CNNs equivariant to very large groups, which grow linearly in computational complexity in the number of dimensions of the group (as opposed to exponentially). This allows us to create G-CNNs equivariant to the Sim(2) group. To the best of our knowledge, Separable G-CNNs are the first work in providing equivariance to this large group.
>
> To the best of our knowledge, Lengyel & van Gemert (2021) is the only existing work, which tackles the problem of separability in the group equivariant setting. However, they only consider the p4m and p4 groups considered in  Cohen & Welling (2016). However, this approach only works for discrete groups where the group action on the convolution kernel can be expressed by a permutation of the kernel grid (a highly restrictive condition). Our work presents a flexible framework for separable group equivariance that generalises to arbitrary compact and non-compact affine Lie groups.
>
> Moreover, we find some considerable differences between depthwise separable convolutions (Haase & Amthor, 2020; Chollet, 2016) and our work. Depthwise separable convolutions essentially separate (a) information mappings in the domain of the data (combining information on spatial configurations of features) from (b) information mappings in the signal itself (combining different features at a single spatial location). separable group convolution kernels, on the other hand, separate information mappings in the domain of the data by (1) first combining information on features under different poses (orientations in case of SE(2)) at a given spatial location, and (2) combining observed features at different spatial locations next. Contrary to Haase & Amthor (2020), Chollet (2016), these convolution operations perform information mappings that split the domain of the data, i.e., the group, to improve computational performance. In addition, we would like to emphasize that ensuring group equivariance while separating its domain is a non-trivial problem, and our method provides a general framework for this purpose.
>
> > 2. The theoretical foundation of this paper is not solid. It is worth mentioning that the authors explored arbitrary affine Lie groups (three different groups: SE2, dilation +translation, and dilation +Sim(2)). But as far as the reviewer knows, this is not entirely applicable. For example, correlation is an equivariant map for the translation but NOT for the rotation group. Why can the authors directly consider SE(2) and Sim (2) in convolution (both containing Rotation)? In addition, when proving translation equivariant, it is necessary to extend f and k to group G, otherwise g^(-1)x does not belong to group G and cannot go on. The reviewer has not seen relevant explanations.
>
> Please note that several works exist which extend the convolution operation to be equivariant to several groups. To name a few: Weiler & Cesa (2019) extends equivariance to E(2) transformations, Satorras et al. (2021) does so for E(n) transformations, Romero et al. (2020), Sosnovik (2019) do so for the scaling group in 1D and 2D.
>
> Nevertheless, no work exists which simultaneously provides equivariance to Sim(2), due to the scalability issue mentioned previously. This particular goal: providing equivariance to large groups in a computational efficient way is precisely the focus and contribution of our work.

---

> > ### Author Response · Authors · 2021-11-18
> > **Response to reviewer 2 (part 3)**
> >
> >
> > > 4. The authors used SIREN as kernel parameterization in the Lie algebra. But as far as the reviewer knows, this work proves the superiority of using periodic activation functions (such as sine), which is, however, irrelevant to this paper from the perspective of the reviewer. Much more details should be provided here.
> >
> > Indeed, our choice of parameterisation wasn’t explained clearly enough in our first submission. We have changed this now. In our original submission, we failed to discuss the increased expressivity that sine activations bring. Sine activations or similar high-frequency inducing functions applied to the data allow MLPs to learn high-frequency functions on low-dimensional domains, where ReLU’s have been shown to underperform (Sitzmann, 2020, (Tancik, 2020). We expanded on section 4 (Defining convolution kernels on Lie algebras), to include these details. Furthermore, we added an experiment to the appendix comparing performance using SIRENs as kernel parameterisation versus using MLPs with ReLU, LeakyReLU or SiLU, which reinforces findings by Romero with regards to kernel expressivity.
> > Lastly, our original submission only briefly mentioned another property of SIRENs that makes them a desirable kernel parameterisation for these group convolution kernels; their explicit control over the smoothness of the learned function through $\omega_0$. We added a section to the appendix visually explaining the importance of kernel smoothness in G-CNNs for continuous groups.
> >
> > > 5. Another concern of the reviewer is that evaluation results of this paper are not convincing to the reviewer. Traditional CNN-based methods have achieved 95+% accuracy on CIFAR10, and regular GNNs (p4m & Z2) can also reach 90+%. The accuracy results of this paper are still inferior to 90% with a 77% baseline. It violates the claim that "G-CNNs usually improve upon regular CNNs" stated in this paper.
> >
> > Unfortunately, we did not motivate the selection of our baseline models properly. This will be changed in our revision.
> >
> > To isolate the effect of separating the group convolution kernels, we intentionally selected a simple architecture for our experiments. Note that this allows us to fairly assess the impact of sampling resolution over the subgroup $H$. Our baseline is achieved with this same architecture, only equivariant to translation (making it equivalent to a regular CNN parameterised with SIRENs). In our experiments, we decisively conclude that equivariance to groups larger than translation (yielding models equating to regular CNNs) does indeed increase performance.
> >
> > Indeed, better architectures for the CIFAR10 task exist. However, they usually rely on several optimization and data tricks, e.g. numerous augmentation methods such as autoaugment, and sophisticated learning rate schedulers, e.g., cosine schedulers, which could potentially hamper the assessment of our work with regard to the separation of the  group convolution. Nevertheless, we appreciate the suggestion to include additional works in our comparison. This provides a better  representation of the capabilities of our models.
> > In particular, we add the following: we include results for an additional experiment in which we achieve SOTA on rotated MNIST with our implementation of $\mathrm{Sim(2)}$-CNNs, considerably improving upon previous results by Weiler et al. (2019), with minimal changes to our experimental setup. We feel this result supports our findings and strengthens our claims about the viability and relevance of our approach to G-CNNs.
> >
> > *edited*: Furthermore, we reimplemented the All-CNN-C architecture used in Cohen & Welling (2016) for CIFAR10, using our separable group convolutions as drop-in replacement for the regular convolutional layers. We show that our $\mathrm{Sim(2)}$ convolutional layers improve performance of the original All-CNN-C, and all versions of the G-CNN proposed by Cohen & Welling (2016), on a similar parameter budget.
> >
> > > 6.  Some parts are not novel at all, such as the section “Depthwise separability”. The reviewer suggests the authors to include more relevant references (Xception or more).
> >
> > We understand the reviewers' concerns with regards to this section. Although we discuss the concept of depthwise separability in section 2 (Separable filters in machine learning), we would like to note that the section Depthwise separability in section 4, has a different purpose. It’s purpose is to illustrate the practical implementation we used in our experiments. We acknowledge that this section could be improved. We have now rewritten this section, and changed its title to better illustrate its purpose.

---

> > > ### Author Response · Authors · 2021-11-18
> > > **References for response to reviewer 2**
> > >
> > >
> > > _(Weiler, M., & Cesa, G. 2019). General $ E (2) $-Equivariant Steerable CNNs. arXiv preprint arXiv:1911.08251._
> > >
> > > _(Sitzmann, V., Martel, J., Bergman, A., Lindell, D., & Wetzstein, G. 2020). “Implicit neural representations with periodic activation functions”. NIPS’20_
> > >
> > > _(Tancik, M., Srinivasan, P. P., Mildenhall, B., Fridovich-Keil, S., Raghavan, N., Singhal, U., ... & Ng, R. 2020). Fourier features let networks learn high frequency functions in low dimensional domains._
> > >
> > > _(Cohen, T., & Welling, M., 2016)  “Group equivariant convolutional networks” ICML’16_
> > >
> > > _(Haase, D., & Amthor, M. 2020). “Rethinking depthwise separable convolutions: How intra-kernel correlations lead to improved MobileNets”. CVPR’20_
> > >
> > > _(Lengyel, A., & van Gemert, J. C. 2021). Exploiting Learned Symmetries in Group Equivariant Convolutions. ICIP’21._
> > >
> > > _(Sosnovik, Ivan, Michał Szmaja, and Arnold Smeulders.2019) "Scale-equivariant steerable networks." arXiv preprint arXiv:1910.11093._
> > >
> > > _(Romero, D. W., Bekkers, E. J., Tomczak, J. M., & Hoogendoorn, M. 2020). Wavelet networks: Scale equivariant learning from raw waveforms. arXiv preprint arXiv:2006.05259._
> > >
> > > _(Satorras, Victor Garcia, Emiel Hoogeboom, and Max Welling. 2021) "E (n) equivariant graph neural networks." arXiv preprint arXiv:2102.09844 ._
> > >
> > > _(Finzi, Marc, Max Welling, and Andrew Gordon Wilson. 2021) "A Practical Method for Constructing Equivariant Multilayer Perceptrons for Arbitrary Matrix Groups." arXiv preprint arXiv:2104.09459 ._
> > >
> > > _(Finzi, M., Stanton, S., Izmailov, P., & Wilson, A. G. 2020). “Generalizing convolutional neural networks for equivariance to lie groups on arbitrary continuous data”. ICML’20._
> > >
> > > _(Bekkers, E. J. 2019). “B-spline cnns on lie groups”. ICLR’20_
> > >
> > > _(Chollet, F. 2017). “Xception: Deep learning with depthwise separable convolutions.” CVPR’17_

---

> > ### Author Response · Authors · 2021-11-18
> > **Response to reviewer 2 (part 2)**
> >
> > >Regarding Sim(2), it has a totally different Lie algebra expression from SE(2)’s due to the scaling, and there has not existed a discussion on specific forms.
> >
> > Importantly, we implemented separable group convolutions on Sim(2). As we are defining convolution kernels over subgroups of Sim(2), this allows us to decouple the logarithmic map used in our kernel parameterisation per subgroup. Since the dilation group has no action on the rotation group, and vice versa, we can simply sample over each subgroup separately, apply the logarithmic map per subgroup, and subsequently expand into a grid over the product subgroup $\mathbb{R}^+ \rtimes \mathrm{SO(2)}$. We understand that these implementational details are important for reproducibility, and will include them in the appendix.
> >
> > > 3. The motivation of the proposed method comes from parameter redundancy, but from Figure 1, the reviewer found that the metric is F-test (variance ratio), which is usually used to analyze statistical models that use more than one parameter to determine whether all or part of the parameters are suitable for estimating the population. And the robustness of F-test is extremely dependent on the samples with a normal distribution. In the case of limited samples, the distribution is often biased, which tends to reduce the power of such a test, leading to incorrect statistical inference.
> >
> > The reviewer raises a valid concern, this experiment should have been explained more clearly. Since no details were given on the number of convolution kernels at this point, from the text it isn't clear whether this experiment is statistically significant. We’ve added details pertaining to the number of spatial kernels we perform PCA on for this experiment to figure 2. Since we treat the spatial kernel at each group element for an input channel for every kernel in a group convolutional layer as a sample, the number of samples we perform PCA on is large (between 16k and 64k). We therefore deem this experiment significant in showing redundancy along the group axis of our learned group convolution kernels. Note that  a similar experiment was conducted by Haase & Amthor (2020) to motivate their approach to inverse depthwise separable convolutions. Lengyel & Van Gemert (2021) motivate their approach in a similar manner by separating the convolution operation found in the original G-CNN by (Cohen, 2016).

---

### Official Review · Reviewer_hC39 · 2021-11-05

**Correctness:** 4
**Technical Novelty And Significance:** 3
**Empirical Novelty And Significance:** 3
**Recommendation:** 8
**Confidence:** 3

**Main Review:**

The strengths of the paper are as follow :

1/ Clarity of the theoretical and empirical parts

2/  Experimental results are promising

3/ The use of SIREN  to parametrize convolutional kernel is a tricky idea and quite original since it is originally proposed for coordinates space data. As a result, a given layer has a fixed number of parameters regardless the resolution of data

4/ Approximating the group convolution integral through random sampling shows a clear advantage over discretization schemes. It allows to mitigate some artifacts of raw data

The weaknesses are as follow :

1/ The proposed model is quite generic. In order to show its scalability, one may wonder if the proposed model could be extended to large scale vision tasks : 3D rendering, Video classification and also to other domains like physical processes where group of symmetries are important.

2/ Lack of comparison with state-of-the art models

3/ Need a further theoretical study


Small typos in appendix, before equation (9) "the group group"



**Summary Of The Paper:**

The paper adresses redundancy in group convolutional filters and the scalability of group ConvNets. It is achieved by introducing a separable group convolution for Affine Lie group which allows to share the kernels for translation elements.

**Summary Of The Review:**

Following the aforementioned consideration, l recommend to accept the paper. The proposed research direction is promising and could open perspectives to several domains, mainly physical process tasks.

---

> ### Author Response · Authors · 2021-11-18
> **Response to reviewer 1**
>
> Dear reviewer 1, we thank you for your insightful review, and for supporting our work. We share your sentiment on the importance of this research direction and indeed, hope this work inspires others to look into scalable solutions to G-CNNs. We will now address your concerns.
>
>
> > The proposed model is quite generic. In order to show its scalability, one may wonder if the proposed model could be extended to large scale vision tasks : 3D rendering, Video classification and also to other domains like physical processes where group of symmetries are important.
>
> We fully agree with the reviewer! The main motivation behind this approach was the exploitation of redundancy to tackle scalability issues present in current approaches to regular G-CNNs. Our experiments were intended to validate this approach, but were by no means exhaustive in the range of possible applications of separable group convolutions. Indeed, the scalability of equivariant approach becomes increasingly relevant as the dimensionality of the data and transformation groups increase, and so we conjecture that separable group convolutions or similar approaches may even be more relevant in settings where data with larger domains than $\mathbb{R}^2$ is involved, where equivariance to transformation groups has proven to be a relevant inductive bias.
> To investigate your suggestion, we are currently working on implementing separable SO(3)-CNNs and applying them to molecular data, e.g., the QM9 dataset. **edit: See the follow-up comment for details!**.
>
>
> > Lack of comparison with state-of-the art models
>
> As the main goal of our experiments is isolating the impact of separating the group convolution operation, we deliberately chose to perform all experiments using the same simple ResNet architecture. As we fully understand the relevance of comparing this approach, in terms of performance, also to other equivariant models, we added a number of experiments. First, we show that with a small modification to our architecture, our Sim(2)-CNN achieves SOTA on rotated MNIST, and we compare and contrast our approach with the previous SOTA by Weiler & Cesa, (2019). We feel this result supports our findings and strengthens our claims about the viability and relevance of our approach.
>
> *edited*: Second, we reimplemented the All-CNN-C architecture used in Cohen & Welling (2016) for CIFAR10, using our separable group convolutions as drop-in replacement for the regular convolutional layers. We show that our $\mathrm{Sim(2)}$ convolutional layers improve performance of the original All-CNN-C, and all versions of the G-CNN proposed by Cohen & Welling (2016), on a similar parameter budget.
>
> > Need a further theoretical study
>
> Could you please elaborate on this? What aspect of the theory would you like to see explored further?
>
>
> > Small typo’s in the appendix
>
> Thanks for pointing these out! We addressed these.
>
>
> _References_
>
> _(Weiler & Cesa, 2019), "General E(2) - Equivariant Steerable CNNs", NeurIPS'19._
>
> _(Cohen, T., & Welling, M., 2016)  “Group equivariant convolutional networks” ICML’16_

---

> > ### Author Response · Authors · 2021-11-22
> > **Follow-up - applying separable G-CNNs to higher dimensional domains**
> >
> > We have implemented a basic ResNet type point convolution architecture for molecular property prediction on the QM9 dataset and ran 3 versions of it: (i) a standard R3 network (Conv3D), (ii) an R3 network with isotropic kernels such that rotation equivariance is guaranteed, (iii) a fully SE(3) equivariant network via separable group convolutions. The net has a point-wise embedding layer, 3 resnet blocks ([conv + batch-norm + act_fn]x2 + skip), and a pooling with prediction layer. It thus 6 convolution layers in total.
> >
> > Considering the time constraints, we have only managed so far to run the experiments for predicting the alpha value, and only trained for 250 epochs. The results confirm what we expect, namely that equivariance constraints are important and are easily implemented with our method. As is standard, we evaluate using the mean absolute error MAE on the predicted property:
> > - R3 Convs: MAE = 0.310
> > - R3 Convs (isotropic kernels): MAE = 0.215
> > - G-Convs (SE(3) equivariant): MAE = 0.142
> >
> > Since these results are quite preliminary and could benefit from more parameter tuning and architecture searches, as well as a full evaluation on all properties, we would like to include these results and accompanying code as a use case in our GitHub repository, rather than including it in the paper as formal results. If the reviewers deem appropriate, however, we would still be happy to include in the paper as preliminary results in an appendix, referring to it in the main body as extra empirical evidence in favor of our work.
> >
> > It was precisely the separability and the flexibility of working with continuous kernel parametrization that allowed us to easily extend the work to 3D translations + rotations. The separability was particularly important in the QM9 point cloud type dataset, as there graph NN had to be used to handle the varying neighborhoods. We could not fit a naïve full group conv implementation in memory (the neighborhoods get too large in the non-sep case), whereas this was no problem for the separable implementation. We worked with random, pseudo-uniform grids of 12 rotations in SO(3), obtained by maximizing the geodesic distance between rotation matrices in the grid.
> >
> > Finally, regarding the significance of these numbers. Our preliminary results are in the ball-park of methods such as tensor field networks (MAE=0.223, Thomas et al.) and SE(3)-transformers (MAE=0.142, Fuchs et al.). The state-of-the-art has predictions in the range MAE 0.04-0.07. We expect that with further tuning we could push our results towards SOTA, but given the timeline we were very happy to see that with minimal effort we could already significantly improve on classic point conv approaches (note early works such as Schnet (Kristof et al.) also use isotropic 3D convolutions like our case (ii) with which they got to an MAE=0.235).
> >
> > _references_
> >
> > _Thomas, N., Smidt, T., Kearnes, S., Yang, L., Li, L., Kohlhoff, K., & Riley, P. (2018). Tensor field networks: Rotation-and translation-equivariant neural networks for 3d point clouds. arXiv preprint arXiv:1802.08219._
> >
> > _Fuchs, F. B., Worrall, D. E., Fischer, V., & Welling, M. (2020). Se (3)-transformers: 3d roto-translation equivariant attention networks. arXiv preprint arXiv:2006.10503._
> >
> > _Schütt, KT, Sauceda, HE, Kindermans, PJ, Tkatchenko, A., & Müller, KR (2018). Schnet - a deep learning architecture for molecules and materials. The Journal of Chemical Physics , 148 (24), 241722._

---

### Author Response · Authors · 2021-11-22
**Changelist**

**General**

- Fixed several typos in the appendix (pointed out by **R1**)
- In concurrence with **R2, R3**, we hope this paper may be helpful as a research guide into the field of scalable equivariant deep learning. To strengthen our work in this regard, we added visualisations of the operations used in (separable) G-CNNs for Lie groups.

**Sec 4. Separable Group Convolutions on Lie Groups**
- We added details on the number of sampled kernels for the experiment creating Fig. 2 (the experimental motivation for separable group convolutions) to address concerns about the statistical significance of these results (**R2**).
- We partially rewrote section **Depthwise separability** and changed its title to **Channel support of separable group convolution kernels**, to better reflect its purpose (**R2**).
- To address questions about the impact of random sampling (**R3**) and our choice of only random sampling over compact groups (**R4**) we expanded on the subsection **Approximating equivariance for compact continuous and non-compact continuous groups** to include these details.
- Expanded on the subsection **Defining convolution kernels on Lie algebras**, to better motivate our use of SIRENs as parameterisation (**R2, R3**).
**Sec 5. Experiments**
- To address concerns on the lack of performance of our approach, and lack of comparison with related work (**R1, R2, R4**), we added two new experiments:
  - An experiment showing we achieve state of the art on rotated MNIST with ${\rm Sim(2)}$-CNNs. We compare these results with a range of baselines proposed in related works.
  - An experiment showing we significantly improve performance of the All-CNN-C (Springenberg, 2014) and all original G-CNNs proposed by Cohen (2016) on CIFAR10 by using our separable Sim(2) convolutional layer as drop-in replacement, with fewer parameters.
**Appendix**
- Partially rewrote section **A.1 Group Theory** to fix some notational mistakes.
- To improve reproducibility (**R2**), we added **A.3 Examples of Lie Groups** detailing the groups we used in our experiments, including implementational details such as the logarithmic maps for these groups.
- To add to the readability of our paper, we add a group-theoretic interpretation of CNNs and figures for operations in (separable) G-CNNs to section **A.4 Visualisations of operations in G-CNNs**
- To further motivate our use of SIRENs (**R2, R3**), we added **A.6 Kernel Smoothness for Group Convolutions on Continuous Groups**, showing the relevance of control over the smoothness of sampled kernels, given by SIRENs.

- To address concerns on lack of experimental details (**R4**), we expanded on appendix **B. Additional Experimental Details**:
  - We expanded the subsection **SIREN architecture and kernel parameterisation**, to more clearly indicate our parameterisation of separable group convolution kernels on the Lie algebra.
  - We added a section **Model Sizes**, showing the number of parameters (**R4**) of our non-separable and separable implementations used throughout our experiments.
  - We expanded on subsection **B.2 Experimental Details** to include further details on the datasets and experimental setups used in our experiments (**R4**).

- We added an additional ablation with $\mathbb{R}^2\rtimes \mathbb{R}^+$-CNNs on CIFAR10 to **C.2 Support of the separable convolution kernel over the channel dimensions**, to address concerns on the impact of additional depthwise separability on the results obtained in our experiments (**R4**), showing how additional depthwise separability outperforms only group separability on a fixed parameter budget.
- To further address concerns about the SIRENs used (**R3**), we added an additional ablation over the hidden size of the SIREN used in **C.3 How does SIREN architecture influence performance?**, showing how performance saturates for larger SIREN hidden sizes and motivating our choice of SIREN architecture.
- To address concerns about the impact of using sine activations in our parameterisation (**R2, R4**), we added an ablation in **C.4 Comparing activation functions for continuous convolution kernels**, comparing performance when using different activation functions in our parameterisation of group convolution kernels, showing superiority of the sine activation.
- To address questions about our choice of only random sampling over compact groups (**R4**), we added an ablation in **C.5 Random sampling over non-compact groups** . show how random sampling significantly impedes model performance of dilation equivariant models.
- **R3** asked an interesting question regarding the redundancy that motivates our separation of the group convolution, and whether it still exists in separable G-CNNs. We added an experiment showing that redundancy decreases over the training process of separable G-CNNs in **C.6 Do separable group convolutions reduce redundancy in G-CNNs?**

---

> ### Author Response · Authors · 2021-11-22
> **Future directions: applying separable group convolutions in higher-dimensional domains**
>
> To further show the relevance of our approach in higher dimensional problems, and answer questions regarding application of our method to other domains (**R1, R4**), we implemented a separable SE(3)-CNN to experiment with molecular property prediction on the QM9 dataset. Although we did not include these experiments in this paper (we leave this for future work), preliminary results are promising, see details in our follow-up responses to **R1, R4**.

---

### Author Response · Authors · 2021-11-22
**General response**

We thank all reviewers for their thorough investigations of our work, and for investing the time to write out valuable insights and criticisms.

We feel the quality of our work has greatly improved through the incorporation of your suggestions, for which we are sincerely grateful to you. Although all reviewers agreed that our submission represents interesting and valuable work in addressing major concerns on the scalability of existing G-CNN approaches to larger groups and data domains, some concerns existed on the originality and experimental assessment of our proposed method. We addressed these concerns by contrasting our approach with existing work and re-emphasizing the need for scalable approaches to G-CNNs. Furthermore, we greatly expanded our experimental investigation into all components of separable group convolutions, and added a number of experiments comparing with related works. With exciting results; with our general framework (which, as we would like to emphasize, can be implemented for arbitrary affine Lie groups) we achieve state of the art performance on the rotated MNIST dataset, the standard benchmark for rotation-equivariant models, and improve performance on CIFAR10 compared to the original All-CNN-C (Springenberg, 2014) and G-CNN (Cohen, 2016) when using our separable group convolution layers as drop-in replacement.

We share the sentiment of reviewers 2 and 3, and hope this work may serve as a research guide for people new to the field of equivariant neural networks. (To this end, we also added visualisations for several group convolution operations in our new submission)

Below we briefly summarize general changes to our submission. We addressed each of your concerns individually as response to your review.

_references_

_Springenberg, J. T., Dosovitskiy, A., Brox, T., & Riedmiller, M. (2014). Striving for simplicity: The all convolutional net. arXiv preprint arXiv:1412.6806._

_Cohen, T., & Welling, M. (2016, June). Group equivariant convolutional networks. In International conference on machine learning (pp. 2990-2999). PMLR._

---

### Decision · Program_Chairs · 2022-01-20

**Decision:**

Reject

**Comment:**

The authors study separable convolutions in the group-convolutional setting, and describe experiments showing them to be more computationally efficient without loss of performance in the setting of some group-augmented MNISTs, and show some promising results on un-augmented CIFAR10, CIFAR100, and Galaxy10.  The reviewers are mixed; some of the reviewers have concerns about the completeness of the experiments and the novelty of the work, and in particular to what extent the experiments support the specific novelties claimed.  The authors have made some updates to address this in the revision, but my opinion is that the authors should resubmit to the next venue after further experiments and exposition to clarify.